# Untangling competition between epitaxial strain and growth stress through examination of variations in local oxidation

**Maria S. Yankova** [1] ✉, **Alistair Garner**[1], **Felicity Baxter**[1], **Samuel Armson**[1], **Christopher P. Race**[1,2], **Michael Preuss**[1,3] ✉ **& Philipp Frankel**[1,2]

Understanding corrosion mechanisms is of importance for reducing the global cost of corrosion. While the properties of engineering components are considered at a macroscopic scale, corrosion occurs at micro or nano scale and is influenced by local microstructural variations inherent to engineering alloys. However, studying such complex microstructures that involve multiple length scales requires a multitude of advanced experimental procedures. Here, we present a method using correlated electron microscopy techniques over a range of length scales, combined with crystallographic modelling, to provide understanding of the competing mechanisms that control the waterside corrosion of zirconium alloys. We present evidence for a competition between epitaxial strain and growth stress, which depends on the orientation of the substrate leading to local variations in oxide microstructure and thus protectiveness. This leads to the possibility of tailoring substrate crystallographic textures to promote stress driven, well-oriented protective oxides, and so to improving corrosion performance.

Due to a spontaneous electrochemical reaction with the environment, most metals corrode by forming a metal-oxide film. This oxide film may protect the metal substrate against further oxidation depending on a combination of numerous competing mechanisms that operate at a range of length scales. Corrosion costs have been estimated to be 3–4% of each nation's gross domestic product or the equivalent of US\$ 2.5 trillion globally in 2013[1]. Hence, there is a financial incentive to reduce corrosion through a better understanding of corrosion mechanisms. For example, slight improvements in the waterside corrosion performance of zirconium alloys, used to encapsulate nuclear fuel, can dramatically improve efficiencies of nuclear power generation[2]. To date, our understanding of the interplay between metal-oxide crystallographic orientation relationships and oxide growth stresses (and the effect of this interplay on the protectiveness of the oxide) is limited due to experimental challenges when analysing highly stressed, nanocrystalline oxides over a statistically significant large area or volume. Although bulk techniques such as X-ray diffraction (XRD) provide good statistics, they commonly lack the required resolution to directly relate oxide texture to specific metal grain orientations or can only be used in single crystal examinations[3–6]. Alternatively, transmission electron microscopy (TEM) investigations[7–9] only focus on a few oxide grains formed on a single randomly selected substrate grain, making a statistical analysis of possible orientation relationships time-consuming and difficult. By definition, these techniques also require the production of electron transparent samples, which is known to alter the microstructure of such stress-stabilised oxide films[10,11].

Studies of both polycrystalline materials[12–14] and single crystals[5,15] have shown that corrosion properties can vary over a few degrees difference in crystallographic orientation. The main effect has been attributed to the metallic dissolution rates, expected to scale directly with the surface energy of the crystallographic planes—close-packed

[1]Department of Materials, the University of Manchester, Manchester M13 9PL, UK. [2]Henry Royce Institute, Royce Hub Building, the University of Manchester, Manchester M13 9PL, UK. [3]Department of Materials Science & Engineering, Monash University, Clayton 3800 VIC, Australia. ✉e-mail: maria.yankova@manchester.ac.uk; michael.preuss@manchester.ac.uk

planes have a higher atomic density and binding energy, and thus show superior corrosion resistance. This trend was confirmed in experimental studies in a wide range of metals—body-centred-cubic materials, such as Fe[12] and Cr[16]; face-centred-cubic materials, such as Ni[3,4,17], Ni-based alloy 22[18], Cu[19], and Inconel 600[20]; and hexagonal-close-packed materials, such as Mg[14], Zn[21], Ti[22], and Zr[5,9,23,24]. In parallel, calculations using first-principles[25–27] and empirical-potential[28–30] methods confirmed the above hypothesis by calculating the surface energies and energy barriers for diffusion along the different crystal directions.

The crystallographic orientation of the metal grain may also directly affect various aspects of the oxide microstructure—such as crystallographic orientation, oxide phases, grain size and shape. The first is a result of lattice matching between the metal substrate and the growing oxide—that is a preferential adoption of an orientation relationship, or epitaxial relationship, between two lattices, where interfacial planes share similar atomic spacing[31]. Epitaxial strains develop, which drive the growth of specific crystallographic texture components in the oxide, thus affecting the lattice and grain boundary resistance to diffusion of corroding species. Cathcart et al.[3] used XRD to measure the epitaxial orientation relationships in the Ni-NiO system for various faces of single Ni grains. They found a correlation between the fraction of high-angle grain boundaries in NiO and the protectiveness of the oxide layer, later confirmed by other studies[4,17]. In the dual-phase Zr-2.5%Nb alloys with a well-defined substrate texture, bulk XRD oxide texture measurements established lattice matching between different metal faces and the $ZrO_2$ film during oxide nucleation, followed by a preferential growth of specific oxide orientations[32]. In contrast, similar XRD measurements on single-phase Zr alloys which exhibit a different 'split-basal' texture[33] did not show strong oxide texture or effects on corrosion kinetics[34]. This observed absence of the effect is an artefact of measuring the macrotexture of the material and, as we will show here, metal-oxide orientation relationships do in fact form in single-phase Zr alloys.

Another factor is the volume change associated with the metal-to-oxide transformation, also referred to as the Pilling-Bedworth ratio[35]. Protective oxides require a ratio larger than one, i.e., the volume of the elementary cell of a metal-oxide is larger than that of the corresponding metal, although Pilling-Bedworth ratios larger than two can be less protective due to oxide buckling[35,36]. Example of ratios between 1 and 2 are Cu, Ni, Zn, Ti, and Zr. Notably, the Pilling-Bedworth-ratio 'rule' only applies to those metal-oxide systems in which the oxygen ion diffuses faster through the oxide than the metal ion, resulting in inwards oxide formation. The $Zr-ZrO_2$ system is a typical example with a Pilling-Bedworth ratio of 1.56 and inwards corrosion—the metal grows a passivating oxide film at ambient temperatures and a semi-passivating oxide layer in typical light-water nuclear reactor conditions[37]. Associated with the volume expansion of a protective oxide is the build-up of compressive stresses, which can be in the range of several GPa[38,39]. These compressive stresses impact oxide grain growth, and thereby grain nucleation, as the oxide will grow so as to minimise its biaxial or triaxial stress state[40,41]. When we consider the influence of the volume change together with the orientation and the elastic anisotropy of the metal crystal, it is possible for oxide orientations, which maximise the accommodation of volume changes normal to the interface, to grow preferentially. During corrosion of zirconium alloys exposed to light-water reactor conditions, an initial layer of small equiaxed oxide grains is formed, followed by the growth of columnar oxide grains of specific crystallographic orientations under the influence of these growth stresses. The growth stresses may stabilise otherwise metastable phases of the oxide, in addition to other stabilising factors such as small grain size and oxygen vacancies[37,42]. For such conditions, the metastable hexagonal sub-oxide ZrO[43] as well as metastable tetragonal $ZrO_2$ along with the stable monoclinic $ZrO_2$ are commonly observed using bulk XRD[37,44–47] and TEM[7–9,48]. There has

been much debate in the literature as to whether tetragonal zirconia is formed as a precursor to monoclinic, or if both can form independently. The oxidation kinetics exhibit a periodicity, which has been related to the oxide microstructure. The oxide layer is protective up to a thickness of about 2 microns, at which point it becomes unstable and a breakdown in the protectiveness of the oxide is observed. This leads to the rapid growth of a fresh protective oxide layer and the process repeats in a cyclic manner. The build-up and subsequent release of the high compressive stresses, as the oxide grains move away from the metal-oxide interface, is proposed to lead to the breakdown of the columnar microstructure and the occurrence of phase transformations between the metastable and stable oxide phases[10,42–44,49–51]. The role of these oxide phase transformations in the corrosion process is still unclear—the majority of studies suggest they have detrimental effects by causing cracking of the oxide film[39,42,44], although a few report beneficial effects[51].

In the present study, we exploit recent technological advances in electron microscopy, in particular, electron backscatter diffraction (EBSD) at low accelerating voltage and scanning precession electron diffraction (SPED) in the TEM. Idealised crystallographic orientation relationships deduced from modelling are directly compared with the orientation of a large number of zirconium oxide grains on different underlying metal grains measured on a bulk sample of Zircaloy-2, corroded for 46 days at 350 °C in simulated pressurised water reactor (PWR) chemistry. Importantly, the bulk sample maintains the constraint in the oxide during the measurement and allows for direct correlation of oxide to substrate grain orientations over a statistically significant area. We present a model for zirconium oxide formation, which resolves the effects of the transformation stress and the epitaxial strain mechanisms on the oxide texture development and oxide phase stabilisation. Our aim is to demonstrate how this multiscale analysis can be used to provide an understanding of the driving forces for oxide texture and microstructure evolution during aqueous corrosion that is applicable to alloys with inwards corrosion and a Pilling-Bedworth ratio between 1 and 2.

## Results
### Electron backscatter diffraction
In order to investigate the protective oxide grown on a commercial single-phase Zr alloy (Zircaloy-2) in a simulated pressurised water reactor environment, the non-protective outer oxide was removed by mechanical polishing (removing ~400 nm from an average initial thickness of ~1.2 μm) and an EBSD orientation map was acquired from the polished oxide surface. The oxide studied is representative of the protective 'pre-transition' oxide, as seen in Fig. 1a, where the transition refers to the cyclic breakdown in the protectiveness of Zr alloys[7]. The use of a low accelerating voltage minimised the electron interaction volume and the resulting map shows that the nanosized oxide grains are grouped into regions with similar crystallographic texture, as outlined and numbered in the figure. Due to the 100-nm step size relative to the average oxide grain diameter of 40–60 nm when viewed in this orientation[8,41], each data point most likely corresponds to a unique oxide grain. We have high confidence in the indexing of the Kikuchi patterns both in terms of orientation accuracy and in phase accuracy, as the monoclinic and tetragonal phases have significantly different crystal structures. The average mean angular deviations (MADs), which indicate if multiple grains are being sampled, are 1.07 and 1.45 for the monoclinic and the tetragonal phase, respectively, and so there would be minimum overlapping in the case of the monoclinic phase and some small overlapping for the tetragonal phase. The accuracy of the indexing using bulk EBSD is further reflected by the similarities in observed textures to those observed by non-destructive XRD measurements in the literature[44–47]. As a result, we obtained a map containing approximately 560,000 oxide grains with 95% of indexed points being stable monoclinic $ZrO_2$ (m-$ZrO_2$) and 5% being metastable

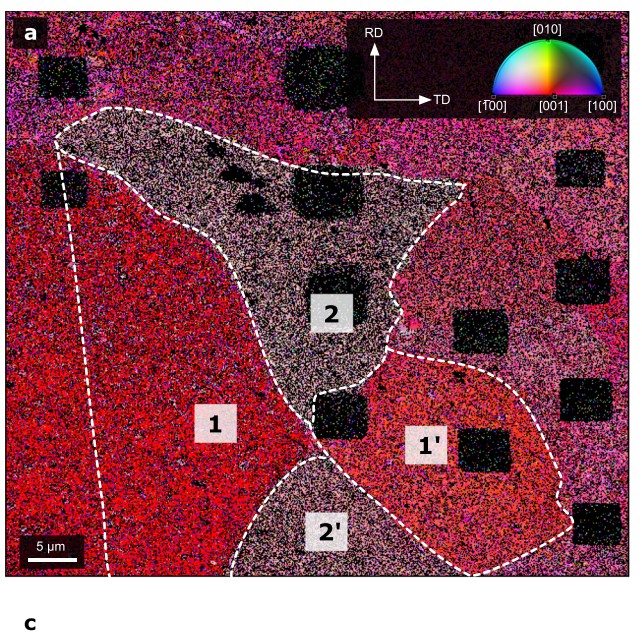

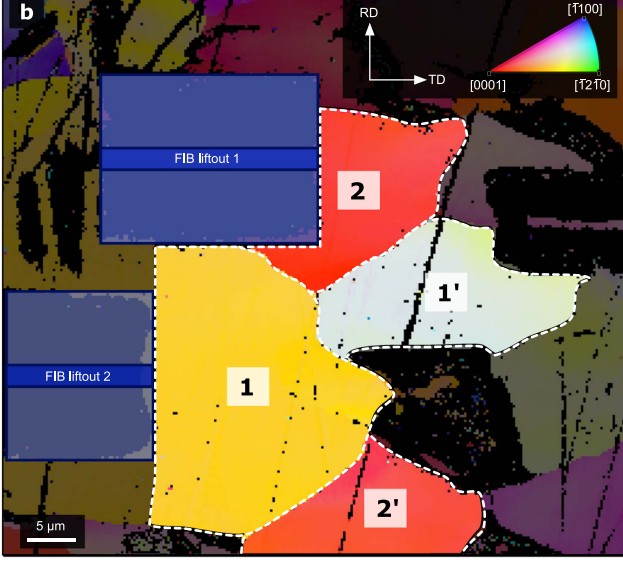

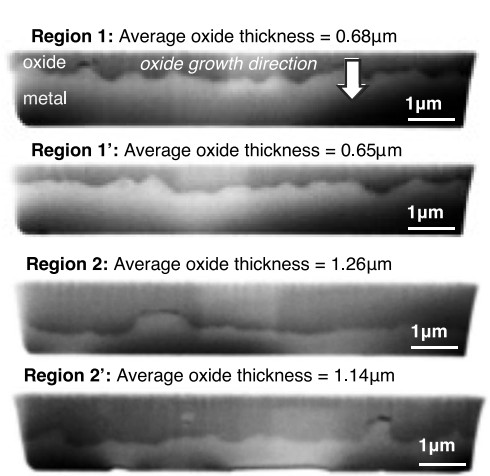

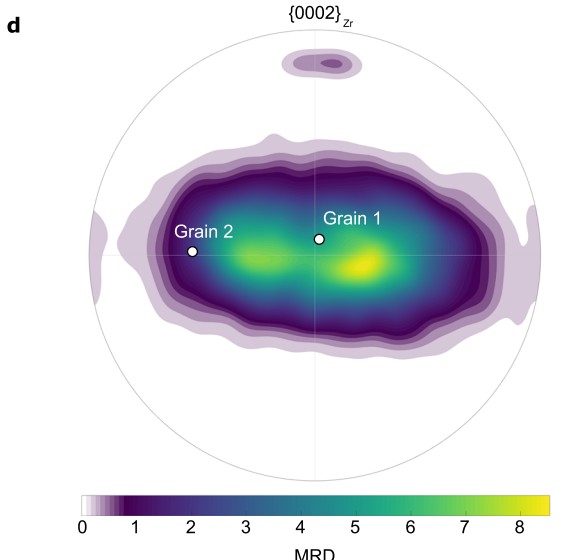

**Fig. 1 | Electron backscatter diffraction orientation maps.** EBSD orientation maps in inverse pole figure colouring relative to ND for **a** monoclinic $ZrO_2$ and **b** corresponding metal grains after mechanical removal of oxide. Oxide regions and metal grains for subsequent orientation analysis are labelled as 1, 2, 1' and 2'. Original rolling (RD) and transverse (TD) directions are marked on the figures. FIB liftout positions for SPED analysis from regions 1 and 2 are shown in **b**. SEM images acquired from FIB trenches from each oxide region are shown in **c**, where the oxide growth direction has been shown for region 1 and is the same for the other regions. Square regions (~5 μm) visible in the oxide orientation map are regions of intentional focused-ion-beam (FIB) damage that formed part of another study[11] and are excluded from this analysis. **d** $<0002>_{Zr}$ pole figure showing the typical 'split-basal' texture of single-phase Zr alloys (Zircaloy-4) measured using electron backscatter diffraction (EBSD), with the orientations of metal grains 1 and 2 marked. Colours represent intensity in units of MRD (multiples of a random distribution).

tetragonal $ZrO_2$ (t-$ZrO_2$) (stabilised by a combination of small grain size[42], stress[52] and oxygen vacancies[53]). Monoclinic $ZrO_2$ is the stable phase of zirconia at temperatures up to 1170 °C with four $ZrO_2$ units per unit cell[54], whereas between 1170 °C and 2370 °C the body-centred tetragonal (bct) $ZrO_2$ phase with two $ZrO_2$ units per unit cell is stabilised. This crystal structure can also be described by a non-primitive face-centred tetragonal (fct) unit cell containing four $ZrO_2$ units, and so comparison between orientations is based on bct Miller indices of the tetragonal cell and equivalent fct indices of the monoclinic cell.

A grain orientation map acquired from the Zr substrate after mechanical removal of the remaining oxide is shown in Fig. 1b. The low temperature stable α-Zr phase has a hexagonal-close-packed crystal structure and hereinafter we refer to that phase as simply Zr. Upon comparison with Fig. 1a, there is a strong suggestion that the borders of the microtextured oxide areas are related to the orientation of the underlying metal grains. The regions/grains are highlighted

accordingly in Fig. 1a, b. The lack of a perfect match can be explained by the difficulty in identifying the exact point at which the metal-oxide interface is reached during removal of the oxide film. We have specifically targeted regions of clearly different oxide textures to understand what produces these. Examining the underlying substrate shows that metal grains 1 and 2 have their basal pole $<0002>_{Zr}$ close to parallel and at about 46°to the normal direction, respectively. These are denoted on the pole figure in Fig. 1d, showing a typical 'split-basal' texture of single-phase Zr alloys[33,55]. This texture is called 'split-basal' due to a symmetrical 'split' of the hcp basal pole with respect to ND that occurs during thermomechanical processing[33]. These alloys also contain a considerable number of grains with the $<0002>_{Zr}$ pole close to parallel to ND (about 20% of the poles are within 15° of ND for this particular sample), which have much thicker oxide. In Fig. 1c cross-sectional SEM images acquired from FIB trenches from each oxide region show the oxide in regions 2 and 2' to be twice as thick as that in

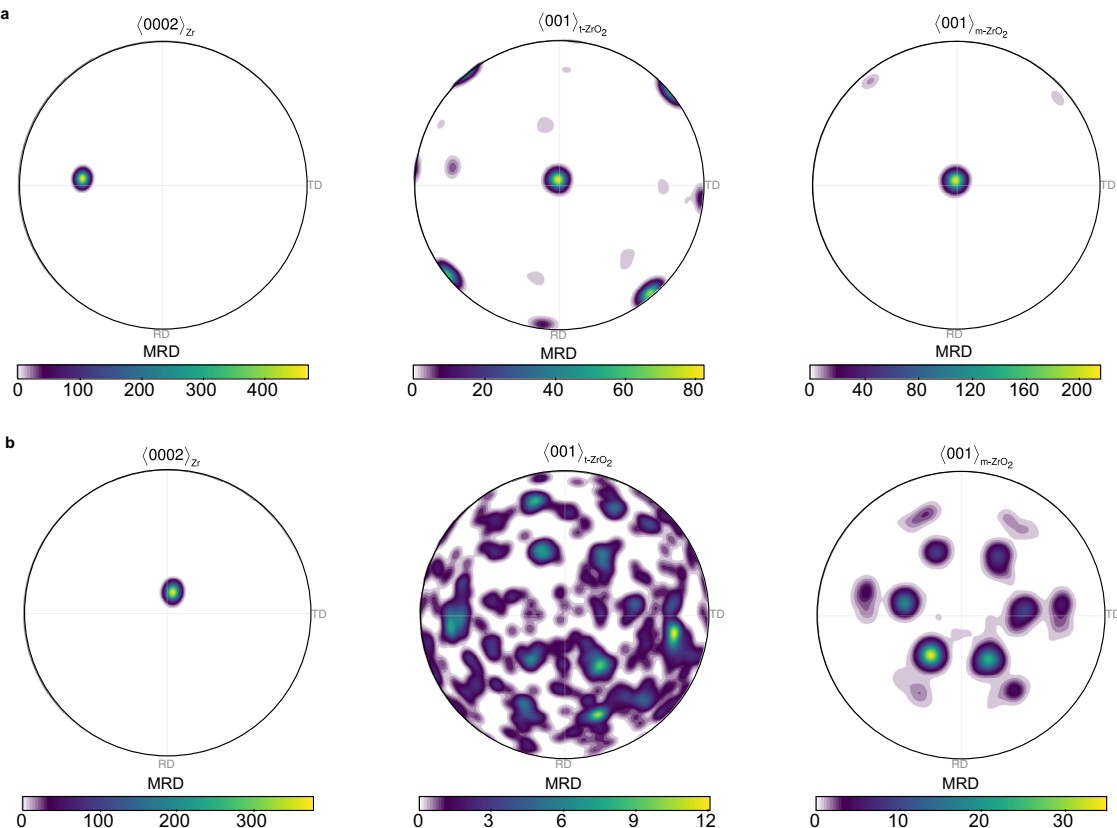

**Fig. 2 | Electron backscatter diffraction contoured pole figures.** Contoured pole figures for the <0002> pole in hexagonal Zr, and the <001> poles in tetragonal and monoclinic $ZrO_2$ in **a** metal grain 1 and oxide region 1; **b** metal grain 2 and oxide region 2. All contoured pole figures are normalised to multiples of a random distribution (MRD) and oriented to be consistent with the EBSD map in Fig. 1 with TD || $x$ axis, RD || $y$ axis and ND || $z$ axis. Colours represent intensity in units of MRD (multiples of a random distribution).

regions 1 and 1'. Further analysis comparing oxide microstructure and crystallographic texture was performed on oxide regions 1 and 2, and the corresponding metal grains. Data from oxide regions 1' and 2' were found to agree with those from regions 1 and 2, respectively, and results are included in Supplementary Fig. 1 and 2.

Firstly, we consider the crystallographic orientation of the metal grains and oxide regions as measured using EBSD in the contour pole figures in Fig. 2. As seen in Fig. 1a, the Zr hcp crystal of grain 1 is positioned so that the basal pole $<0002>_{Zr}$ is inclined at ~46° to the normal direction (ND). In oxide region 1, we measured 49,938 monoclinic grains and 1847 tetragonal grains, or a tetragonal phase fraction of 3.6%. We observe that the <001> poles of the monoclinic and the few indexed tetragonal grains in this region are oriented strongly in the oxide growth direction. The tetragonal <001> also shows four other peaks positioned at 90° with respect to ND. Figure 2b shows that the ⟨0002⟩ pole of the hcp Zr crystal in substrate grain 2 is close to parallel to the normal direction, oriented at 12°. In oxide region 2, the ⟨001⟩ tetragonal $ZrO_2$ pole figure is formed from 336 tetragonal grains, which shows a close-to-random crystallographic texture. Further confidence in the results for the tetragonal texture is given by their confirmation from the comparable regions 1' and 2', formed from 258 and 285 grains respectively, which can be found in Supplementary Figs. 1 and 2. On the other hand, the ⟨001⟩ pole figures, formed from 20,544 measured monoclinic oxide grains, exhibit a six-fold symmetry with stronger preference for two of the texture variants. There is a lower tetragonal phase fraction of 2.5% compared with region 1.

Plotting the raw pole figures from the EBSD map for the tetragonal and the monoclinic $ZrO_2$ phases in region 1, Fig. 3a, c, suggests the presence of lattice matching between the metal substrate phase and the tetragonal and/or the monoclinic oxides. We performed further analysis by calculating possible theoretical orientation relationships, which revealed two scenarios. In the first one, denoted by pink discs in Fig. 3a, c, hcp Zr transforms to tetragonal $ZrO_2$, and then the tetragonal grains transform to monoclinic $ZrO_2$ based on the relationships: $\{111\}<10\bar{1}>_{m-ZrO_2}||\{101\}<11\bar{2}>_{t-ZrO_2}||\{0002\}<11\bar{2}0>_{Zr}$. In the second one, denoted by blue crosses in Fig. 3c, hcp Zr can still transform to tetragonal $ZrO_2$ according to $\{101\}<11\bar{2}>_{t-ZrO_2}||\{0002\}<11\bar{2}0>_{Zr}$, but it can also transform directly to monoclinic $ZrO_2$ according to $\{111\}<10\bar{1}>_{m-ZrO_2}||\{0002\}<11\bar{2}0>_{Zr}$. We observe almost equal proportion of the four experimentally measured texture components in the $\{001\}_{m-ZrO_2}$ and $\{10\bar{1}\}_{m-ZrO_2}$ pole figures, which are matched by the theoretical orientations of the first scenario. Therefore, we conclude that the majority of the oxide has formed as the tetragonal phase first, and then transformed to monoclinic. That resolves a long-standing debate of which $ZrO_2$ phase forms first and has important implications for the corrosion process of Zr alloys to be discussed later. It should also be pointed out that during a hexagonal-close-packed to tetragonal phase transformation, 24 possible symmetrically equivalent variants exist. A crystallographic symmetry variant refers to a crystal with the same crystal structure as another variant, but with a different orientation with respect to the parent crystal from which it formed during a phase transformation. However, the hcp crystal is only 3-fold rotationally symmetric, and so 12 of the symmetry variants might grow on an exposed A layer of the hcp stacking and another 12 might grow on an exposed B layer. We only observed one of these 12-variant sets in the measured tetragonal phase, which we attribute to a random process due to the transformation of a single Zr grain. In the next phase transformation, from tetragonal to monoclinic $ZrO_2$, we note that 4 of

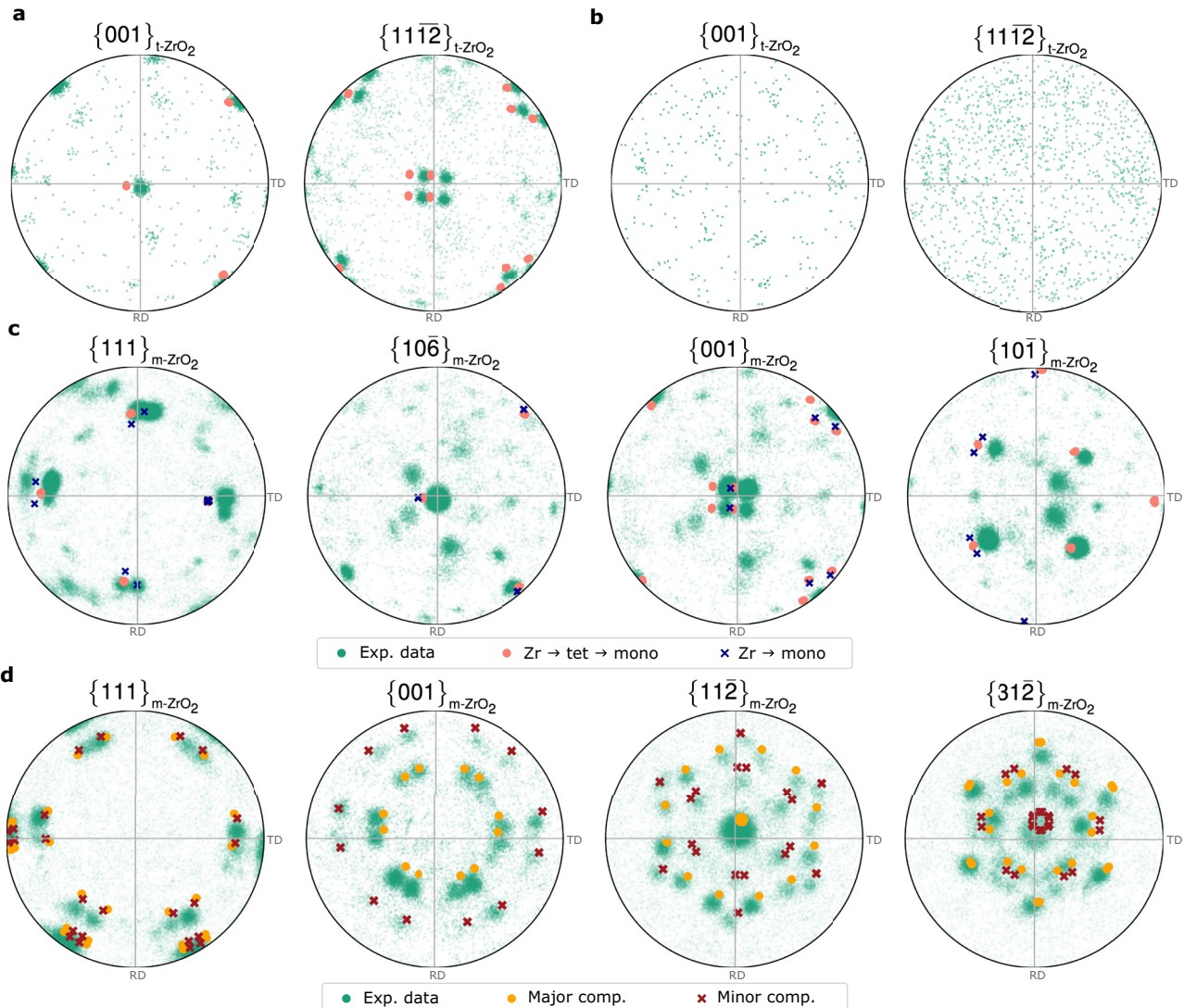

**Fig. 3 | Electron backscatter diffraction and theoretical crystallographic orientation data. a, b** Raw pole figures for equivalent poles {001} and {11$\bar{1}$2} of tetragonal ZrO$_2$ in oxide region 1 (**a**) and 2 (**b**). **c** Raw pole figures of monoclinic ZrO$_2$ for equivalent poles {111}, {10$\bar{6}$}, {001} and {10$\bar{1}$} in oxide region 1. Denoted with pink disks and blue crosses are possible theoretical orientation relationships {111}<10$\bar{1}$>$_{m-ZrO_2}$||{101}<11$\bar{2}$>$_{t-ZrO_2}$||{0002}<11$\bar{2}$0>$_{Zr}$ and {111}<10$\bar{1}$>$_{m-ZrO_2}$||{0002} <11$\bar{2}$0>$_{Zr}$, respectively. **d** Raw pole figures for equivalent poles {111}, {001}, {11$\bar{2}$}

and {31$\bar{2}$} of monoclinic ZrO$_2$ in oxide region 2. Denoted with orange disks and red crosses are the major and minor theoretical orientation relationships {11$\bar{2}$} <111>$_{m-ZrO_2}$||{0002}<1$\bar{1}$00>$_{Zr}$, and {31$\bar{2}$}<111>$_{m-ZrO_2}$||{0002}<1$\bar{1}$00>$_{Zr}$. All contoured pole figured are normalised to multiples of a random distribution (MRD) and oriented to be consistent with the EBSD map in Fig. 1 with TD || $x$ axis, RD || $y$ axis and ND || $z$ axis.

the 12 variants have grown preferentially, as seen in Fig. 3c, to form a strong epitaxial texture with the {10$\bar{6}$}$_{m-ZrO_2}$ parallel to the metal-oxide interface. Raw pole figure data plotted from the EBSD maps of the tetragonal and the monoclinic ZrO$_2$ in region 2 are shown in Fig. 3b, d. Here, the tetragonal grains do not exhibit a preferential texture. On the other hand, the {001}$_{m-ZrO_2}$ pole figure in Fig. 3d shows two distinct texture components. We investigated different possible oxide orientations that could arise from lattice matching between the hcp Zr lattice and the monoclinic ZrO$_2$ lattice, and to reproduce the experimental results the inclusion of two epitaxial relationships between the monoclinic oxide and the metal substrate was required. We have identified both of these components to have the ⟨111⟩ monoclinic oxide directions parallel to the ⟨1$\bar{1}$00⟩ Zr directions but with either the {11$\bar{2}$} or the {31$\bar{2}$} oxide planes parallel to the {0002} metal plane. The theoretical orientations based on these two epitaxial relationships, i.e. {11$\bar{2}$}<111>$_{m-ZrO_2}$||{0002}<1$\bar{1}$00>$_{Zr}$ and {31$\bar{2}$}<111>$_{m-ZrO_2}$||{0002} <1$\bar{1}$00>$_{Zr}$, and the assumption of the metal ⟨0002⟩$_{Zr}$ being normal to the metal-oxide interface, are overlaid as

orange disks and red crosses, respectively. We see an excellent match between the model and the experimental textures. The fraction of the major texture component was found to be approximately 2.4 times that of the minor component and both components have a spread of 16° from the ideal orientation for reasons described previously.

## Scanning precession electron diffraction

A SPED correlation index map for oxide region 1 is shown in Fig. 4a. The correlation index is a measure of the quality of matching between the experimental and the theoretical diffraction patterns from template matching[56] and can be used to visualise microstructural features analogous to band contrast maps produced from EBSD measurements. The map is overlaid with monoclinic grain boundaries defined by a misorientation threshold of 5° to emphasise the oxide microstructure and to enable grain size analysis. In region 1, the oxide grains are well-aligned and exhibit a uniform oxide microstructure with an average grain width of ~57 nm. In Fig. 4b the SPED phase map from region 1 clearly shows that most of the oxide is composed of monoclinic ZrO$_2$

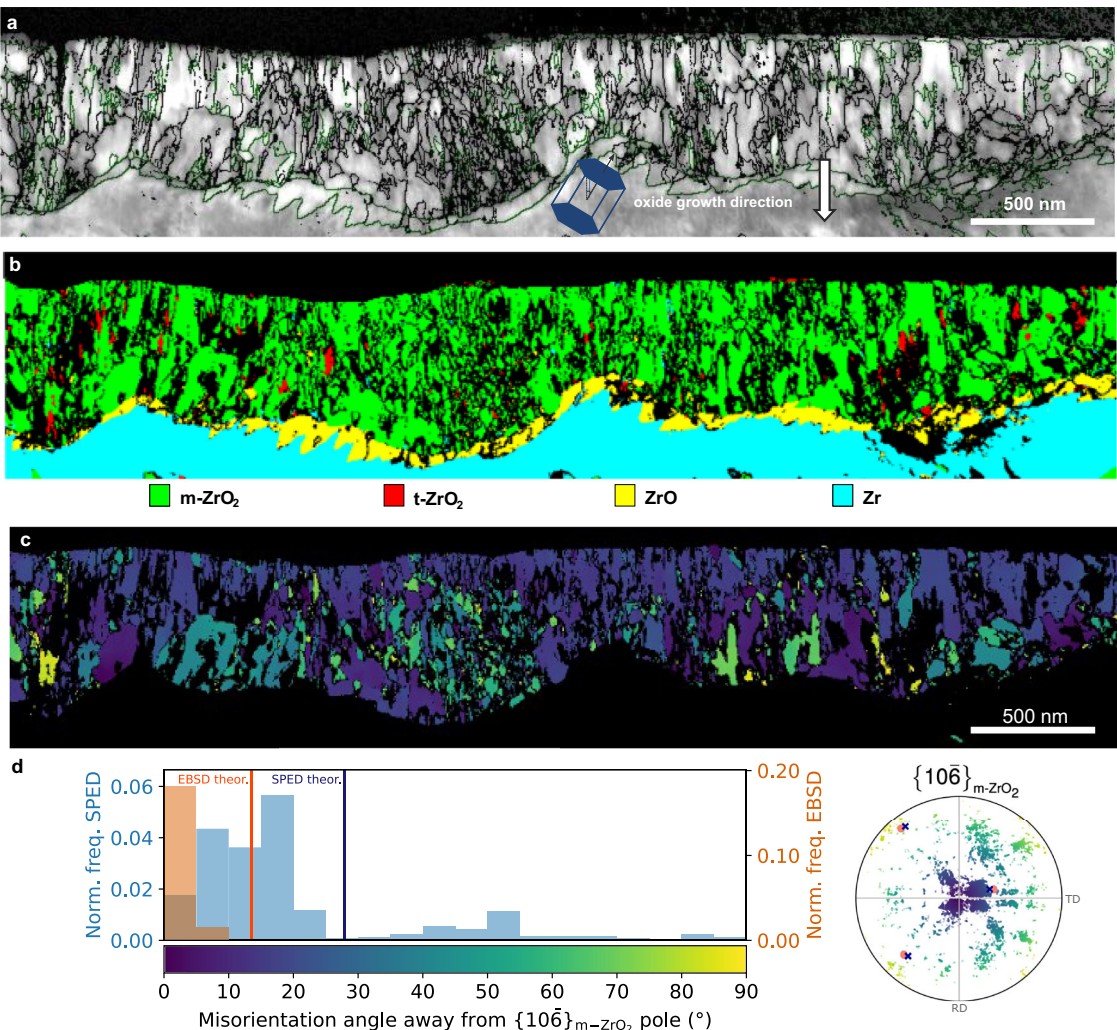

**Fig. 4 | Oxide microstructure in region 1 measured using scanning electron precession diffraction (SPED) in the TEM. a** Correlation index map with overlaid monoclinic oxide grain boundaries shown in black (interphase boundaries shown in green). The orientation of the Zr hcp unit cell is shown and the oxide growth direction is marked with an arrow. **b** Phase map with phase reliability values less than 10 shown as non-indexed points (in black) and indexed points coloured according to legend below (**b**). **c** Monoclinic ZrO$_2$ orientation map coloured according to degrees away from the {10$\bar{6}$} pole according to colour legend in **d**. **d** Normalised frequency of orientations and pole figure with respect to degrees away from the main texture component {10$\bar{6}$}$_{m-ZrO_2}$ for both EBSD and SPED data. Denoted with pink disks and blue crosses are possible theoretical orientation relationships {111}<10$\bar{1}$>$_{m-ZrO_2}$||{101}<11$\bar{2}$>$_{t-ZrO_2}$||{0002}<11$\bar{2}$0>$_{Zr}$ and {111}<10$\bar{1}$>$_{m-ZrO_2}$||{0002}<11$\bar{2}$0>$_{Zr}$, respectively.

with a minor tetragonal phase fraction in this region of around 2% and a continuous layer of ZrO sub-oxide. Figure 4c shows the SPED orientation map of the monoclinic oxide formed in region 1, which is coloured according to deviation of the {10$\bar{6}$}$_{m-ZrO_2}$ pole from the oxide growth direction. As shown in the frequency distribution in Fig. 4d, the SPED measured texture exhibits a slightly larger spread compared with the EBSD measured data. The corresponding theoretical variants with a misorientation angle from the {10$\bar{6}$}$_{m-ZrO_2}$ pole closest to zero are also plotted. The theoretical orientations are calculated based on the measured substrate orientation using the corresponding microscopy technique. Although there is a 15° difference between the maximum of the misorientation angle away from the {10$\bar{6}$}$_{m-ZrO_2}$ distributions of the EBSD and the SPED monoclinic oxide texture measurements, the difference between the experimental maximum and the corresponding theoretical value is about 13° for both techniques. These values confirm a very good agreement between the two techniques and between the experimental measurements and the theoretical model. Several factors contribute to the difference between the EBSD and the SPED measurements, the biggest one being the difference in the sampling statistics in the two techniques. Further factors include

misalignment between the samples used in each technique, internal misorientations within the metal grain, misindexing of some SPED data and removal of constraint when preparing a TEM foil. There is also a difference in the measured grain populations, as EBSD is biased towards measuring larger oxide grains compared with SPED.

Figure 5a shows the SPED correlation index map from within region 2, where the oxide exhibits a complex microstructure consisting of narrow columnar grains and regions of equiaxed nanostructured oxide, with a smaller average grain width of ~41 nm when compared with region 1. The SPED phase map in Fig. 5b shows that the majority of the oxide in this region is also composed of the monoclinic ZrO$_2$ phase. The tetragonal phase fraction is <1% and is mainly confined to small, isolated equiaxed grains. There are also some isolated larger grains at the metal-oxide interface which are indexed as hexagonal ZrO sub-oxide phase in agreement with previous observations[57]. Figure 5c, d show the m-ZrO$_2$ orientation map, coloured as degrees away from the {11$\bar{2}$} main texture component, and the corresponding normalised frequency of orientations compared with that from EBSD. Most of the monoclinic oxide grains are oriented between 0° and 35° away from the {11$\bar{2}$} pole with a peak of

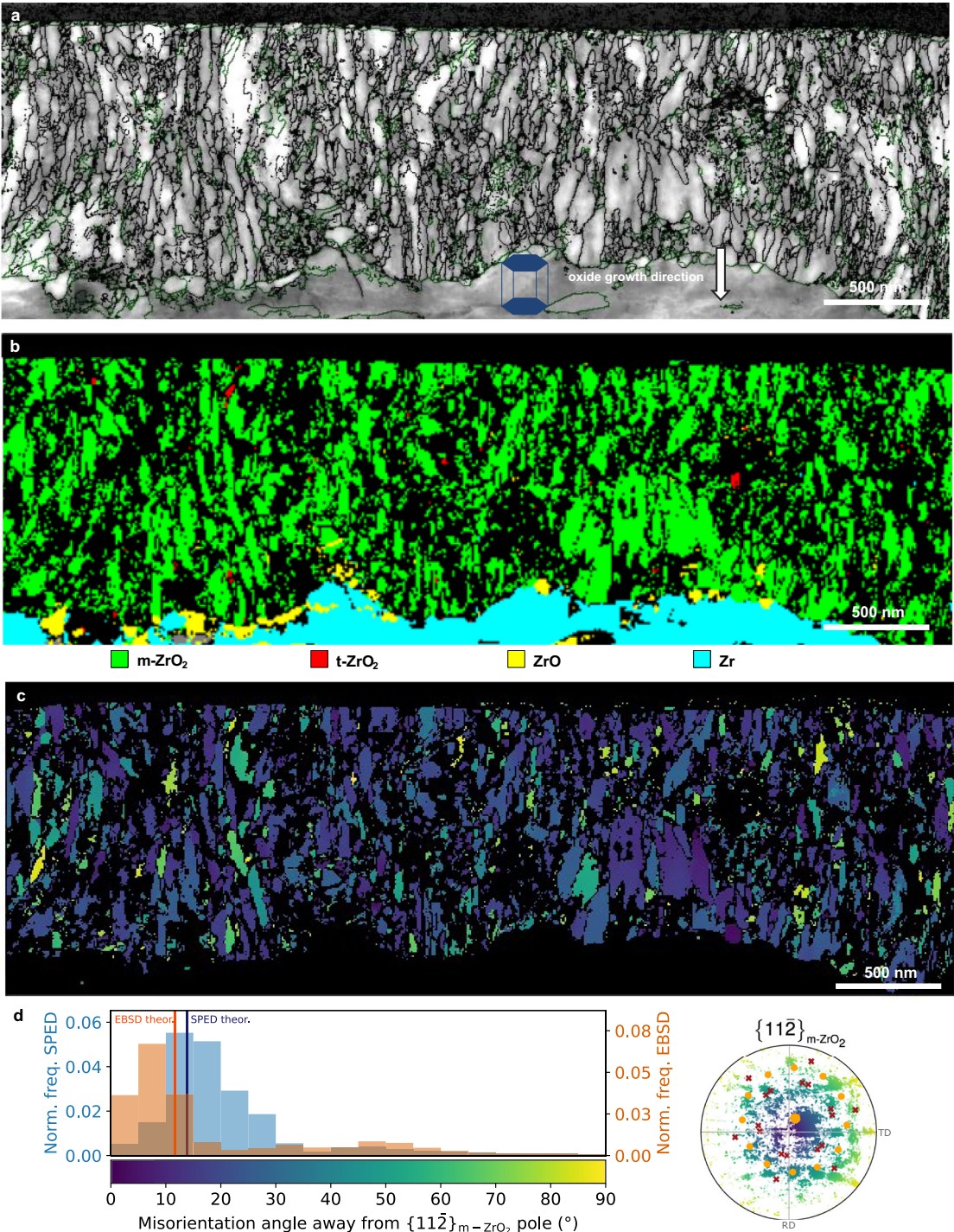

**Fig. 5 | Oxide microstructure in region 2 measured using scanning electron precession diffraction (SPED) in the TEM. a** Correlation index map with overlaid monoclinic oxide grain boundaries shown in black (interphase boundaries shown in green). The orientation of the Zr hcp unit cell is shown. The oxide growth direction is marked with an arrow. **b** Phase map with phase reliability values less than 10 shown as non-indexed points (in black) and indexed coloured according to the legend below (**b**). **c** Monoclinic $ZrO_2$ orientation map coloured according to degrees away from the $\{11\bar{2}\}$ pole according to colour legend in **d. d** Normalised frequency of orientations and pole figure with respect to degrees away from the main texture component $\{11\bar{2}\}_{m-ZrO_2}$ for both EBSD and SPED data. Denoted with orange disks and red crosses are the major and minor theoretical orientation $\{11\bar{2}\}\langle111\rangle_{m-ZrO_2}||\{0002\}\langle1\bar{1}00\rangle_{Zr}$ and $\{31\bar{2}\}\langle111\rangle_{m-ZrO_2}||\{0002\}\langle1\bar{1}00\rangle_{Zr}$.

orientations at about 10°. As expected, the SPED data exhibits a larger spread compared to the EBSD data due to the smaller number of oxide grains sampled. The SPED $\{11\bar{2}\}$ pole figure, also coloured as degrees away from the main texture component $\{11\bar{2}\}$ and overlaid with the theoretical orientations from the orientation relationships calculated based on the orientation of the metal in the SPED data, confirm the presence of these two monoclinic texture components.

We also observe a slightly larger misalignment between the experimental and the theoretical data compared to that in the EBSD measurement, however, we can see the same pattern. The theoretical peaks of the $\{11\bar{2}\}$ are very close to those obtained using both SPED and EBSD as seen in Fig. 5d with differences between the maximum of the misorientation angle distribution and the theoretical peaks of 4° and 7° for, respectively, SPED and EBSD. Additionally, SPED allows us

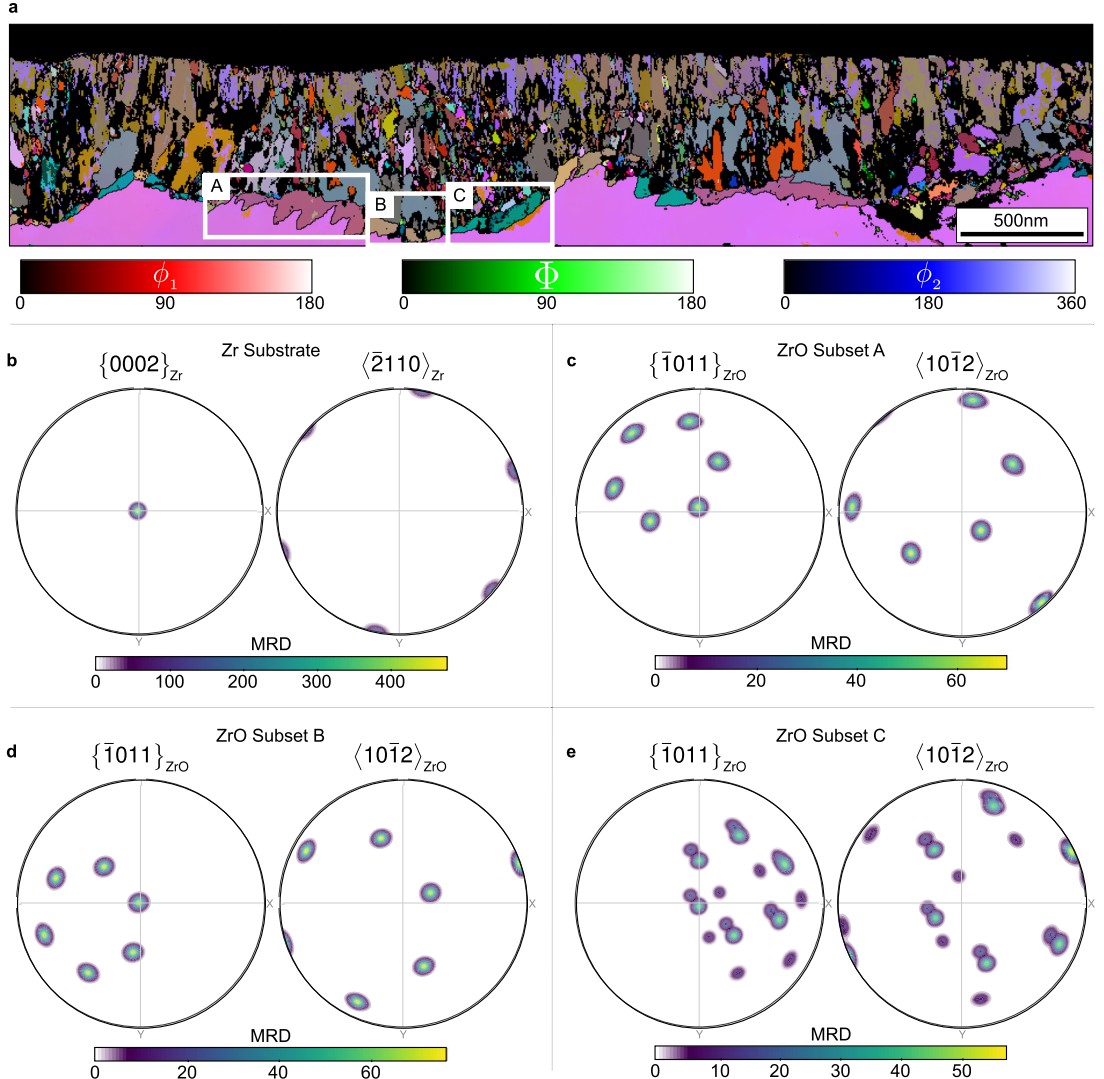

**Fig. 6 | Sub-oxide orientations in region 1 as measured by scanning electron precession diffraction (SPED). a** Orientation map from region 1 in Euler angle colouring as shown in legend with ZrO sub-oxide subsets A, B, and C marked. **b** Contoured {0002} and ⟨2̄110⟩ pole figures for the Zr substrate. **c–e** Contoured {1̄011} and ⟨101̄2⟩ pole figures for the ZrO sub-oxide subsets A, B and C labelled in **a**. The {0002}$_{Zr}$ is aligned parallel to the pole figure z direction to make orientation relationships clearer. Colours represent intensity in units of MRD (multiples of a random distribution).

to investigate the potential role of the ZrO sub-oxide on the oxide texture. In region 1, the sub-oxide forms an almost continuous layer between the metal substrate and the ZrO$_2$ film, demonstrating the characteristic 'sawtooth' sub-oxide morphology[57] as seen in Fig. 4b. On the other hand, fewer isolated grains were observed in region 2. Further analysis showed that an orientation relationship between the metal and the ZrO phase exists in region 1 but not in region 2. Figure 6a shows the SPED orientation map coloured according to Euler angles for all major phases formed in region 1. The oxide has been split into subsets A, B and C in Fig. 6a, which cover the three main sub-oxide orientations observed in this region. Contoured {0002} and ⟨2̄110⟩ pole figures for the Zr substrate and {1̄011} and ⟨101̄2⟩ pole figures for the different ZrO sub-oxide regions are shown in Fig. 6b–e, where the substrate {0002} pole is aligned with the 'z' direction of the pole figure in order to better visualise any orientation relationships. The pole figures show that in subset A and B, the sub-oxide follows the main orientation relationship identified in[50], with {0002}$_{Zr}$||{1̄011}$_{ZrO}$ and ⟨2̄110⟩$_{Zr}$||⟨101̄2⟩$_{ZrO}$. This is also in agreement with TEM observations of sub-oxide formation during in situ annealing[49]. In subset C, the orientation relationship is less clear, as this region is composed of two distinct sub-oxide orientations,

however both of them are also close to the major orientation relationship reported previously[49,50].

## Discussion

Detailed analysis over significant length scales shows that the orientation of Zr grains has a dramatic effect on the subsequent development of the local oxide nanostructure and texture, and thus the local protectiveness of the oxide. This has wide-reaching implications for the understanding of oxide nucleation and growth processes and highlights the importance of multiscale characterisation and correlative microscopy/modelling in understanding these complex processes. Focusing on two oxide regions separated by about 10 mm, identical material and corrosion conditions resulted in marked differences in oxide protectiveness, which seems to be strongly correlated to differences in the orientation of the underlying metal grains.

The schematic in Fig. 7 summarises our findings for the competing mechanisms of oxide nucleation and growth. In a typical metal grain in 'split-basal' textured single-phase Zr alloys, the c-axis of the hcp crystal is positioned at 20° to 40° away from the outer surface normal, and so pyramidal planes with Miller indices {h0il} are close to parallel to the outer surface (Fig. 7a). In contrast, these alloys also

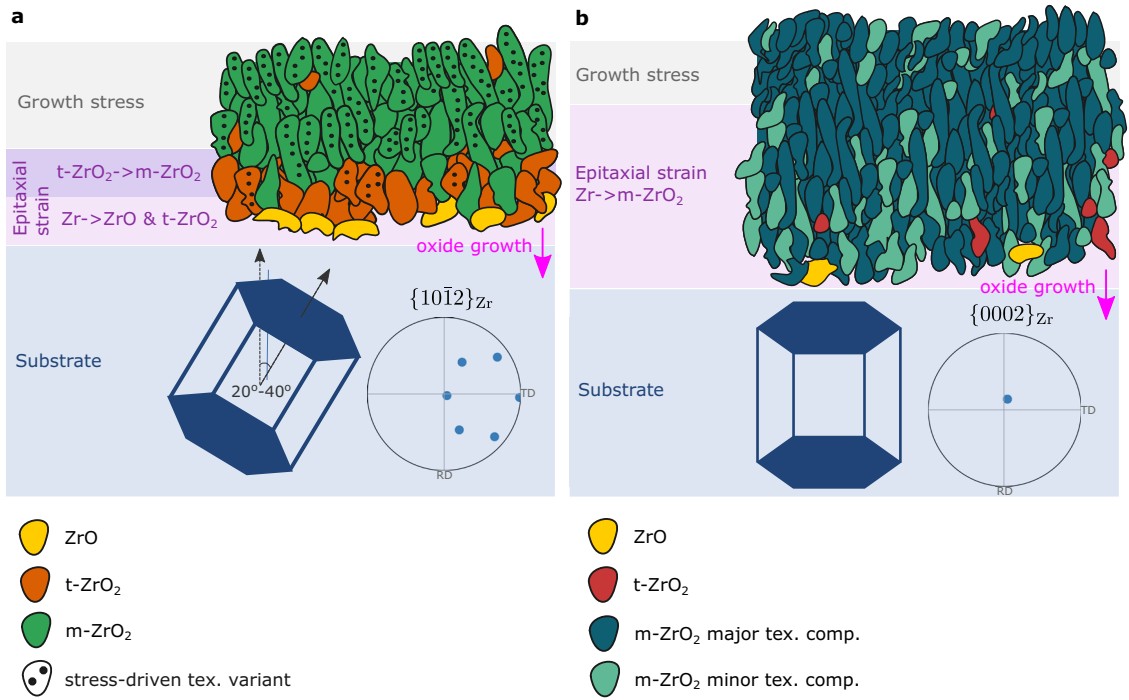

**Fig. 7 | Schematic representation of the mechanisms of Zr metal to oxide transformation and their effect on the grain morphology and texture of the oxide film based on the orientation of the substrate metal grain.** Two types of oxide microstructure are observed based on the orientation of the substrate grain. **a** Region 1: a protective oxide layer with long columnar grains and low energy grain boundaries under the influence of the growth stress. **b** Region 2: less protective oxide, thicker and with more disordered smaller grains. **a** ZrO, t-ZrO$_2$, and m-ZrO$_2$ grains are shown in yellow, orange and green, respectively, and dots indicate stress-driven texture variant; **b** ZrO, t-ZrO$_2$ and m-ZrO$_2$ grains are shown in yellow, orange and green, respectively, where the darker and lighter shade of green indicates the major and minor m-ZrO$_2$ texture components. The oxide growth direction is marked with an arrow.

contain a smaller fraction of grains with basal planes close to parallel to the outer surface, represented in Fig. 7b. According to our observations, the former substrate grain type forms a thinner oxide film with wider and more columnar grains, with higher fraction of metastable ZrO and tetragonal ZrO$_2$ phases, compared with the latter. In both cases, the majority of the oxide consists of monoclinic ZrO$_2$, however, with a strong one-component epitaxial texture in the first case, and a two-component epitaxial texture in the second case.

We now discuss how the crystallographic orientation of the metal grain influences the formation of these oxide microstructures. Since the basal plane is the most densely packed and the lowest energy plane in the hcp Zr lattice, it is expected to be the most corrosion resistant. The pyramidal plane {h0il} orientations are expected to exhibit properties in between those of the basal and the prism orientations as they lie on an energetically favoured region in the surface energy anisotropy projection with a minimum at the basal orientation {0002} and a maximum at the prism orientation {10$\bar{1}$0}, as calculated by a broken-bond based geometric model[58]. Multiple experimental and theoretical studies[5,27,59] on single crystal Zr have shown that the prismatic planes oxidise more readily in water and in pure oxygen than the basal plane with an estimated oxygen diffusion twice as fast along the [10$\bar{1}$0] direction compared with the [0002] direction. Nevertheless, based on our FIB cross-section analysis, the pyramidal orientations that belong to the 'split-basal' texture showed a corrosion rate two times lower than that of the basal-type orientations. A similar result was reported for Ni[18,20], where under certain thermodynamic conditions, the corrosion rate did not follow the surface energy anisotropy of face-centred-cubic metals. Instead, the formation of a passivating oxide film slowed down the corrosion rate on certain surfaces, and the surfaces most susceptible to corrosion were found to be those most able to form a protective film[18]. In the present case, that effect can be attributed to the competition between the epitaxial strain, i.e., the lattice matching, and the growth stress in the nucleation and growth stages of oxide

formation. A charge-optimised many-body potential study by Noord-hoek et al.[30] demonstrated that water dissociates faster, and that atomic O and H diffuse further into the prism planes with a more even distribution compared with the basal plane—87% of O was found to lie between the top two layers for {0002}, compared with 54 and 44% for {11$\bar{2}$0} and {10$\bar{1}$0}, respectively. Therefore, oxygen atoms will be more evenly distributed and penetrate to a greater depth in substrate orientations that fall close to the ideal 'split-basal' texture, such as that in region 1. That would lead to a higher fraction of oxygen vacancies in the same volume of material, and hence, favour the formation of the lower-stoichiometry ZrO phase and the vacancy-stabilised tetragonal ZrO$_2$ phase, when compared with the basal orientation that corresponds to region 2.

Both types of Zr grains oxidise by strong alignment of the child oxide phase with the parent metal phase according to specific epitaxial orientation relationships. In the case of a grain with an orientation close within the peaks of the 'split-basal' texture of Zr alloys, it was previously thought that no orientation relationship exists between the metal and the oxide[34]. We identified that the orientation of the metal grain energetically favours the nucleation of tetragonal ZrO$_2$ based on the epitaxial relationship {101}<11$\bar{2}$>$_{t-ZrO_2}$||{0002}<11$\bar{2}$0>$_{Zr}$, and in some cases hexagonal ZrO based on the epitaxial relationship {$\bar{1}$011}<10$\bar{1}$2>$_{ZrO}$||{0002}<11$\bar{2}$0>$_{Zr}$. And so, when the sub-oxide ZrO phase does form, it can transform to tetragonal ZrO$_2$, maintaining the same tetragonal texture as that of the grains directly nucleated from the Zr metal. Lattice matching also drives the tetragonal to monoclinic phase transformation based on the relationship {111}<10$\bar{1}$>$_{m-ZrO_2}$||{101}<11$\bar{2}$>$_{t-ZrO_2}$. We note that these three orientation relationships can be combined as {111}<10$\bar{1}$>$_{m-ZrO_2}$||{101}<11$\bar{2}$>$_{t-ZrO_2}$||{$\bar{1}$011}<10$\bar{1}$2>$_{ZrO}$||{0002}<11$\bar{2}$0>$_{Zr}$. In contrast, we have shown that a Zr metal grain with the c-axis close to normal to the interface, is more likely to directly nucleate as monoclinic ZrO$_2$, with <111>$_{m-ZrO_2}$||<1$\bar{1}$00>$_{Zr}$ and {11$\bar{2}$}$_{m-ZrO_2}$||{0002}$_{Zr}$ or {31$\bar{2}$}$_{m-ZrO_2}$||{0002}$_{Zr}$.

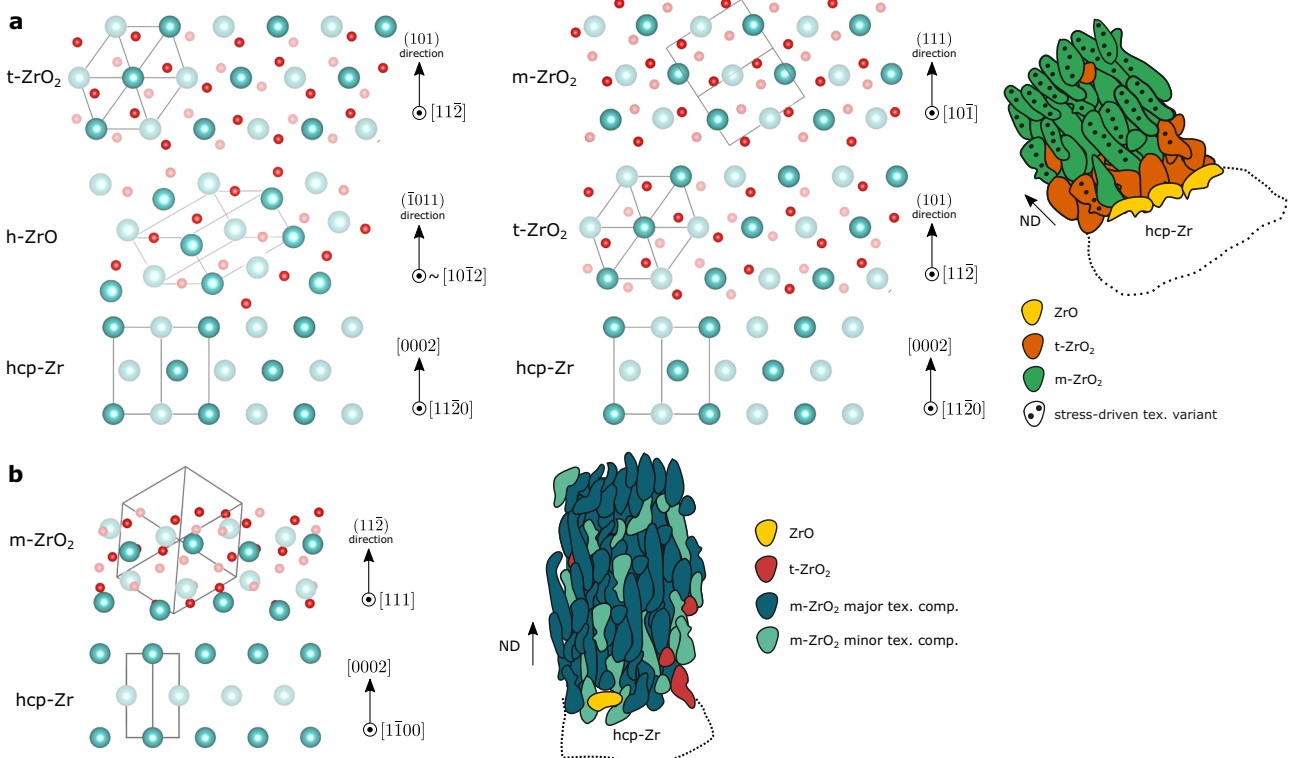

**Fig. 8 | Atomic structure of interface.** Schematic of the interfaces formed based on the identified orientation relationships. **a** Region 1: $\{101\}\langle11\bar{2}\rangle_{t-ZrO_2}||\{\bar{1}011\}$ $\langle10\bar{1}2\rangle_{ZrO}||\{0002\}\langle11\bar{2}0\rangle_{Zr}$ (left) and $\{111\}\langle10\bar{1}\rangle_{m-ZrO_2}||\{101\}\langle11\bar{2}\rangle_{t-ZrO_2}||\{0002\}$ $\langle11\bar{2}0\rangle_{Zr}$ (right). **b** Region 2: $\{11\bar{2}\}\langle111\rangle_{m-ZrO_2}||\{0002\}\langle1\bar{1}00\rangle_{Zr}$. Two atomic planes shown in the out-of-page direction, where shade indicates the atoms' relative distance and darker atoms are closer to the reader. Blue and red spheres represent Zr and O atoms, respectively. Produced using VESTA[69]. Schematics of the oxide microstructure show the normal direction and how it relates to the crystal directions for reference.

Figure 8 shows the atomic structures of the interfaces between the metal and oxide phases in the two types of regions, and Table 1 lists the epitaxial mismatch required between phase transformations. The mismatch values have been calculated using the Zr–Zr distance within the corresponding crystal structure and represent theoretical differences that are used as a qualitative measure of the epitaxial strains in the two regions. We note that there is no difference in the total epitaxial mismatch in the two regions when we consider the transformations from Zr metal to monoclinic $ZrO_2$. Therefore, we consider the effect of growth stresses on the oxidation process, which is determined by a combination of the effect of areal footprint of the symmetry variant and the crystal stiffness anisotropy of the child phase[32,38]. The child phase variants formed from different parent symmetry operators will have different areal footprints on the outer surface based on the orientation of the parent crystal. For example, when the

Zr basal pole is parallel to ND, the symmetry variants would have the same footprint, whereas when it is positioned at an angle to ND, these will differ. The epitaxial strains necessary to form a coherent interface between the two lattices, on the other hand, would stay the same−the atomic interface is the same but rotated with respect to the global (sample) coordinate system. And so, in region 1, we would have an inequivalent distribution of areal footprints and stiffnesses, some of which will be favourable for growth based on minimising the stresses and with most of the volume change accommodated normal to the metal-oxide interface[39]. On the other hand, in region 2, the epitaxial strains remain strong and drive all symmetry variants to grow in equal proportion.

Based on the epitaxial relationships in region 1, the nucleated tetragonal variants will have either the {001} or the {110} plane close to parallel to the outer surface, which then transform to $\{10\bar{6}\}$ and {100} in the monoclinic phase, respectively. Previously, preferential growth of the $\{001\}_{t-ZrO_2}$ and the $\{10\bar{6}\}_{m-ZrO_2}$ was attributed to their small areal footprint as a determining factor that minimises the compressive stress[32]. However, these orientations have very similar areal footprints −26.0 Å² and 26.4 Å² for $\{001\}_{t-ZrO_2}$ and $\{110\}_{t-ZrO_2}$, and 27.3 Å² and 27.9 Å² for $\{10\bar{6}\}_{m-ZrO_2}$ and $\{100\}_{m-ZrO_2}$, respectively. Figure 9a shows the directional dependence of Young's modulus in the body-centred tetragonal $ZrO_2$ crystal, which suggests the $\{001\}_{t-ZrO_2}$ has a lower stiffness compared with $\{110\}_{t-ZrO_2}$. Thus, the growth of the $\{001\}_{t-ZrO_2}$ variants, which subsequently transform to $\{10\bar{6}\}_{m-ZrO_2}$, would minimise the growth stress in the oxide film. These results agree with previous studies, which have reported similar monoclinic oxide textures of $\{10\bar{l}\}$ (where $l = 2, 3, 4$)[46,60]. The particular value of $l$ will depend on the metal grain orientation that the oxide has transformed from. Importantly, our results show that the tetragonal phase is a required precursor to the monoclinic phase, in order for a protective oxide layer

## Table 1 | Epitaxial mismatch between different Zr and oxide phases

| | Phase transformation | In-plane mismatch (%) | Average in-plane mismatch (%) | Average out-of-plane mismatch (%) |
|---|---|---|---|---|
| Reg. 1 | hcp-Zr -> h-ZrO | 6.6, −5.4 | 0.6 | 0.6 |
| | hcp-Zr -> t-ZrO₂ | 10.1, 11.0 | 10.6 | 12.0 |
| | h-ZrO -> t-ZrO₂ | 3.8, 15.5 | 9.7 | 11.5 |
| | t-ZrO₂ -> m-ZrO₂ | - | 1.4 | −0.6 |
| | total hcp-Zr -> m-ZrO₂ | - | 11.7 | 11.5 |
| Reg. 2 | hcp-Zr -> m-ZrO₂ | - | 11.8 | 11.5 |

Calculated based on the difference between the Zr–Zr distances in the corresponding lattices. Shear is ignored.

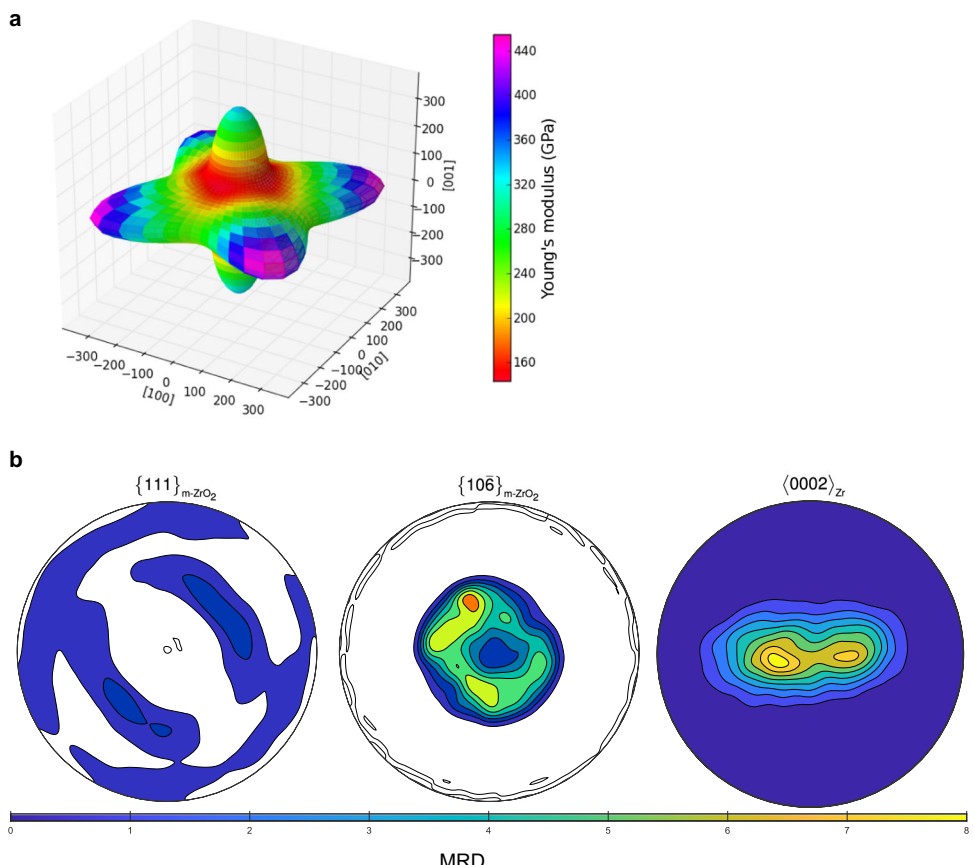

**Fig. 9 | Young's modulus of t-ZrO$_2$ and modelled macrotexture. a** Directional dependence of Young's modulus in the body-centred tetragonal ZrO$_2$ crystal produced using the SC-EMA tool[70]. **b** Contoured pole figures for the {111} and {10$\bar{6}$} monoclinic ZrO$_2$ poles as modelled using the orientation relationships obtained in this study based on Zircaloy-4 substrate with a crystallographic texture shown in the <0002> Zr contour pole figure, as measured using electron backscatter diffraction (EBSD). Colours represent intensity in units of MRD (multiples of a random distribution).

to form with a strong single-footprint epitaxial texture consisting of wider and more columnar grains. Based on the crystallographic orientations, the sub-oxide was found not to be a necessary predecessor of the tetragonal phase. It has been suggested that it might be a byproduct of slow oxidation[8,61]. However, it might also have beneficial effects through a more gradual change in the stoichiometry of the Zr ion, as demonstrated by Ma et al.[62]. These stronger textured oxides are likely to have a higher fraction of protective, low energy grain boundaries, and also coarser, more coherent microstructures as it has previously been postulated that columnar grain growth is terminated by small mismatches between local grain orientations[46].

Previously, the tetragonal to monoclinic phase transformation was thought to cause the onset of cracking and porosity in the oxide layer, and thus has a mainly determinant role on the oxidation process[37]. We need to point out that there are different mechanisms of tetragonal phase stabilisation—chemical stabilisation due to alloying elements forming sub-4+ valence state during corrosion and compressive stress stabilisation. In either case, this stabilisation can be temporary, allowing tetragonal ZrO$_2$ grains to grow beyond their otherwise critical size, as the compressive stresses will reduce as the metal-oxide interface proceeds inwards and an increase in oxygen partial pressure allows partially oxidised alloying elements to fully oxidise[63]. In contrast, tetragonal grains stabilised throughout the oxidation process, whether due to small size or vacancies, are likely to have a beneficial effect by allowing for the formation of a coherent and protective microstructure. Another potential contributing factor to the more protective oxide in region 1 is that the metastable phases allow a more gradual volume expansion from metal to oxide, and a

more gradual change in the stoichiometry[62], and so fewer defects develop in the oxide microstructure. The volume strain necessary to transform Zr to ZrO is 11.9%, the strain to transform ZrO to tetragonal ZrO$_2$ is 29.3% and the expansion from tetragonal to monoclinic ZrO$_2$ is 5.9%, as opposed to a sudden expansion of 53.3% from Zr to monoclinic ZrO$_2$, as occurs in region 2.

Both orientation relationships in region 2 form an oxide with a single unique areal footprint—the {11$\bar{2}$}$_{m-ZrO_2}$ with a footprint of 31.5 Å$^2$ or the {31$\bar{2}$}$_{m-ZrO_2}$ plane with a footprint of 32.1 Å$^2$, since they are both formed from a metal grain with the basal plane normal to the oxide growth direction. The similar footprints, and potentially similar stiffnesses of the two orientations, leads to the epitaxial strain, rather than the growth stress, being the stronger mechanism, driving the growth of both texture components with a higher prevalence of the {11$\bar{2}$}$_{m-ZrO_2}$ orientation. It is the simultaneous growth of two epitaxial texture components in combination with the direct transformation from Zr metal to monoclinic ZrO$_2$ that most likely enhances oxygen ion transport through the oxide as the elongated oxide grains are not well organised and more high energy grain boundaries are formed reducing the level of passivation.

Moreover, we considered the oxide texture that would result due to a transformation of a large number of metal grains present in a typical sample with 'split-basal' texture. The results are shown in Fig. 9b, where all metal orientations with a basal pole within 20° of ND were assumed to transform according to the orientation relationships in region 2, and the remaining metal orientations were assumed to transform according to the orientation relationships in region 1 with a preferential growth of the oxide orientations closest to {10$\bar{6}$}$_{m-ZrO_2}$.

The monoclinic oxide {111} contour pole figure shows a close to uniform distribution, which agrees well with previous macrotexture measurements[34,45,46]. The small differences between the theoretically calculated and the experimentally measured pole figures, such as the position and spread of the peaks, can be attributed to differences in the 'split-basal' texture of the metal that the oxide has formed from, and the number of oxide grains considered. Combined with the result for the $\{10\bar{6}\}_{m-ZrO_2}$ pole figure, it can be seen how the macrotexture of the oxide appears to have a preferential fibre. However, this macrotexture is formed by a large number of epitaxial textured oxide grains formed from metal grains with different orientations. Our study demonstrates that there are epitaxial orientation relationships determining the oxide orientations, and it is incorrect to describe the oxide crystallographic texture as a 'fibre' texture.

In summary, we demonstrate the importance of substrate orientation on the protectiveness of the oxide against corrosion and the complex interplay between the two mechanisms of epitaxial strain and growth stress on that process. In alloys with inward corrosion and a Pilling-Bedworth ratio of more than one, epitaxial strain due to lattice matching between the substrate and the oxide drives the growth of a more disorderly nanostructured oxide. In contrast, the growth stress encourages a more well-ordered, coarser-grained oxide microstructure and leads to improved global corrosion performance. This finding opens the possibility for improved corrosion performance of a large range of alloys through process optimisation avoiding specific grain orientations even if they are a minority texture since they can still result in local reduction of passivation.

## Methods

### Material selection
The material used in this study was sheet Zircaloy-2 material in the recrystallised condition with an average equiaxed grain diameter of ~35.9 μm and whose chemical composition lies within the range defined in ASTM B353, with 1.5 wt% Sn, 0.14 wt% Fe, 0.1 wt% Cr and 0.06 wt% Ni, where the balance is Zr[64]. The coupon was subjected to corrosion testing for 46 days at 350 °C ± 0.5 °C in a 316H stainless steel autoclave in simulated PWR chemistry at a raised pH level as part of a large-scale testing programme[65], forming an average oxide thickness of ~1.2 μm as estimated from weight gain data. Zirconium forms a protective adherent oxide film, where all the oxygen in the chemical reaction produces zirconium oxide, therefore, the weight gain of the specimens is typically used as a direct measure of the oxide film thickness[45,66].

### Sample preparation
In order to reduce the topography inherent to oxidised zirconium alloys and to remove the outer portion of equiaxed grains related to the fast initial oxidation process[41], a 2 × 2 mm section of the oxidised Zircaloy-2 sample was prepared using 4000 grit silicon carbide paper with a final polish in 0.06 μm colloidal silica. A total of ~0.4 μm of oxide was removed, as measured by subsequent cross-sectional measurements. EBSD analysis was performed on this surface as discussed below. Following EBSD analysis, two TEM liftouts were made from different areas of the prepared oxide surface using the in situ liftout technique with an FEI Quanta 3D focused-ion beam (FIB) instrument. These samples were thinned to electron transparency for SPED in the TEM analysis using the standard FIB procedure with a final low-energy cleaning step. The position of these liftouts is shown in Fig. 1b.

To evaluate the local oxide thickness, trenches were prepared in different oxide regions using the same FIB instrument. The oxide surface was firstly protected using Pt, a regular cross-section pattern was then used to mill out a region of the oxide and underlying metal using an accelerating voltage of 30 KeV and a current of 3 nA. A final polish was performed on the cross-sectional surface at 1 nA to provide a clean surface for accurate oxide thickness measurement. Secondary electron imaging was performed at 5 KeV with a current of 1.6 nA.

The remaining oxide was then carefully removed using 0.06 μm colloidal silica. The material removal rate was relatively slow to ensure that significant amounts of the Zr substrate were not removed. Once the oxide was removed, the underlying metal microstructure was assessed with a further EBSD map (details below), using trenches as fiducial markers to match the metal and oxide mapping region.

### Electron backscatter diffraction
In order to obtain reliable bulk EBSD analysis of nanograined materials such as $ZrO_2$, it is vital that the interaction volume of the electron beam is minimised. In the present case this is achieved with a low accelerating voltage of 10 KeV and a current of 1.6 nA using an FEI Magellan 400 XHR Field Emission Gun Scanning Electron Microscope (FEG-SEM). The unique column design on this microscope allows for a high current density to be maintained in a small spot size in order to get sufficient signal out of the sample at low accelerating voltages/currents. The disadvantage of using such beam conditions is that the backscatter signal on the detector is relatively weak. This is overcome using the high-resolution Oxford Instruments Nordlys Nano 2 EBSD detector, operated at ~7 Hz, using 4 × 4 binning. For the oxide measurement, a total area of 50 × 50 μm² was measured with a step size of 100 nm, with a total acquisition time of ~24 hours. The patterns were indexed using the Oxford Instruments Aztec software suite with an indexing rate of ~60%. Due to the small oxide grain size, each indexed point in the map most likely corresponds to a unique oxide grain. It should be pointed out here that there are small ~5 μm squares in the orientation map that appear to show a different crystallographic texture. These are regions of intentional FIB damage that formed part of another study[11], and are excluded from this analysis.

After oxide removal, a second EBSD map was obtained from the underlying metal. Due to the relatively large grain size, an accelerating voltage of 30 keV was used with a current of 3.2 nA using the same microscope and detector set up as used for mapping the oxide. Patterns were acquired with 8 × 8 binning over a total area of 100 × 100 μm. A 250 nm step size was used at a speed of ~70 Hz giving a total acquisition time of 40 minutes. As with the oxide map, the patterns were indexed using the Oxford Instruments Aztec software suite with an indexing rate of ~80%. All EBSD data were analysed using the Channel 5 software suite developed by Oxford Instruments and the open-source MATLAB toolbox MTEX[67]. A similar number of indexed points in all regions were analysed for a more accurate texture comparison.

### Scanning precession electron diffraction
In order to analyse the local microstructure and microtexture in different oxide regions in more detail, SPED was performed on the two FIB liftouts from regions 1 and 2 in Fig. 1b. In this technique, an electron diffraction pattern is recorded by scanning a small, parallel electron probe over the sample and then automated diffraction indexing algorithms are used to match the recorded diffraction patterns against theoretically derived templates in order to simultaneously determine both the phase and the orientation[56]. SPED phase and orientation maps were acquired using a Tecnai TF30 transmission electron microscope operated at 300 KeV fitted with a Nanomegas ASTAR SPED system. The patterns were acquired at 25 frames per second with a step size of 5 nm and a precession angle of 0.4 degrees and indexed using the ASTAR template matching procedure[56]. Orientations and phases identified with a reliability index >10 were excluded from the analysis, as that is the optimum value for Zr oxide[68]. The grain size is estimated using the linear intercept method from the resultant grain boundary maps, with grain boundaries defined by a critical misorientation threshold of 5°. Although the analysis of the entire oxide thickness is not possible due

to the removal of the outer portion of oxide, an accurate measurement of the grain widths in the inner oxide can be performed.

## Oxide texture modelling

A Python code was developed in order to model the oxide texture formed on the observed substrate orientations. The raw EBSD orientation data was used to find the theoretical child phase orientations for a given orientation relationship between metal and oxide or between two oxide phases. The epitaxial relationships were found by starting with the substrate orientation, generating a large number of possible oxide orientations by incremental rotation of the oxide lattice in 3° intervals until a match between the theoretical and experimental equal-area-projection pole figures was found. Then, the smallest Miller indices planes for these orientations were calculated within a 10° tolerance. The misalignment between the theoretical and the experimental spots is most likely a result of a combination of factors including error in the orientation measurement of the metal grain (e.g., some distortion in the sample when the oxide is polished off), the undulated interface between the metal and the oxide, and internal misorientation in the metal grain.

## Data availability

All data needed to evaluate the conclusions in the paper are included in the paper and/or the Supplementary Information. The raw EBSD and SPED orientation and phase data may be found at 10.5281/zenodo.4737700. The raw diffraction data will be made available on request.

## Code availability

The Python code used to create the texture models and to explore the experimental data may be found as Jupyter Notebooks at 10.5281/zenodo.4737700.

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

## Acknowledgements

The authors gratefully acknowledge funding from the Engineering and Physical Sciences Research Council UK (EPSRC) through the Centre for Doctoral Training in Advanced Metallic Systems, EP/G036950/1 (M.S.Y., A.G., F.B., S.A.) and MIDAS programme grant EP/S01702X/1 (M.P., C.P.R., P.F.). C.P.R. was funded by a University Research Fellowship of the Royal Society. The authors are thankful to the MUZIC consortium for valuable discussions.

## Author contributions

All authors contributed extensively to the conceptualisation and the design of the methodology of the study. A.G., F.B. and S.A. performed the experiments and the initial processing of the experimental data.

M.S.Y. performed further processing and analysis of the experimental data and created the models. M.S.Y., S.A., A.G. and F.B. created the visualisations. C.P.R., M.P. and P.F. supervised the work. The original draft was written by M.S.Y. and A.G. M.S.Y., A.G., S.A., C.P.R., M.P., and P.F. reviewed and edited the manuscript.

## Competing interests

The authors declare no competing interests.
