## [Peer Review File · Nature Communications]

Untangling competition between epitaxial strain and growth stress through examination of variations in local oxidationREVIEWER COMMENTS

Reviewer #1 (Remarks to the Author):

Detailed TEM examination was carried out on two TEM specimens lifted out of two oxidized Zr grains. It is a long way to expect that these provide representatives for electrochemical corrosion of all alloys in all harsh environments. There are some good insights however presentation could be much better. There appears to be confusion between preferential growth of some oxide grow directions and transformations between oxide types and Zr metal. A major effort to explain the research would be useful. The present version is extremely difficult to understand, and maybe even incomprehensible to a highly trained researcher in the field. The abstract should present main findings and put them into context. There are no conclusions.

Incomprehensible, needs to be translated into language that is understandable to the trained practitioner in oxidation studies

a. Please number pages!

1. The problem definition is vague. Harsh environment. Is this molten salt at 1000 C, salt water, or CO₂ + H₂S at 500 C. Please be a little more specific.

2. Is this electrochemical corrosion in an aqueous NaCl solution or a molten salt?

3. It would be useful to have a clear statement of research aims to replace the long discussion at the end of the Main Text.

4. Full chemical composition should be given.

5. Justification is needed that weight gain is a useful method to determine passive film thickness, and that there is no oxide loss during corrosion.

6. Please explain all acronyms. What is SPD?

7. This work is based on a 2 mm x 2 mm specimen that was oxidised, had most of the oxide removed by grinding, from which two TEM specimens were produced and examined. How representative are these? Justification is needed.

8. These oxides are thick compared with passive films, 1-2 nm in thickness.

9. Oxide growth direction is undefined.

10. the transformation from Zr to ZrO₂ is presumably by oxidation, If so please state this. What is the

evidence for the transformation of tetragonal ZrO₂ to monoclinic ZrO₂, and how does this occur?

11. Or are you trying to say that Zr oxidised to both oxide variants independently? Or are you trying to say that one oxide variant grows on top of the other? What is the evidence for any of this?

12. What is precession diffraction? Please explain. Presumably the black areas are where there is lack of data?

13. It would be good to indicate the orientations of the oxide grains and of the Zr grains in Fig. 4(a)

14. The majority of the grains in Fig. 5b are black which presumably means unidentified

15. The model in Fig. 7 is neat. However, please indicate the confidence level, as most of grains were black in Fig. 5.

16. The presented discussion about water splitting can only be relevant until the first oxide monolayer forms.

17. What is the orientation of a Zr grain with a split basal orientation? There appears to be a mixup of macroscopic and grain focused conceptions.

18. There is also confusing terminology about oxide transformation, when the model indicates growth of grains of particular orientations on other grains.

19. The atomic relations in Fig. 8 are instructive. However, the split-basal orientation needs to be included.

20. Are the strains in Table 1 relevant. If so please provide TEM plane spacings of these oxides in support

Reviewer #2 (Remarks to the Author):

The study investigates the mechanisms contributing to zirconium oxide phase formation during corrosion of zirconium. The research method utilized orientation mapping techniques in scanning electron microscope and transmission electron microscope to determine the texture in parent and oxide phases. The experimental data and the theoretical calculations orientation relationship in the phases reveal the energetic favor of the tetragonal oxide phase nucleation on zirconium substrate during corrosion. This is the first time that a study demonstrates the possibility of tetragonal oxide nucleation in zirconium though such theories have been discussed. The manuscript argues that the orientation of the zirconium substrate would affect the oxide nucleation and growth process. Thus, the research work

argues the influence of epitaxial strain and growth stress in the formation of oxide phases and their texture in zirconium.

The following are the areas where more clarity and information are needed.

1. The authors' conclusion on the nucleation of the tetragonal oxide phases is based on the post-corrosion analysis of the microstructure and assumed orientation relationship calculations between Zr, tetragonal oxide, and monoclinic oxide. The analyzed sample has more than 95% transformed monoclinic oxide phases. Indeed, the results are high-quality. However, the authors may consider using in situ TEM observation of such tetragonal oxide nucleation can provide direct and more convincing evidence. Another option is to have an oxidation condition under which the tetragonal phase is a major phase. There are several papers regarding using precession electron diffraction to study the oxidation of Zr or zircaloy and the tetragonal phase is popular (J. Nucl. Mater. 556 (2021) 153196; Scr Mater. 145 (2018) 95).
2. In region 2, the number of tetragonal oxide grains analyzed is ~530, which is 3-4 times lesser than that in region 1. The pole figures in region 2 show close to random texture. How representative is the pole figure calculated for the phase in region 2 from a smaller number of grains?
3. Since each oxide grain has one orientation data point in the SEM-EBSD study, it is critical to comment on the correctness of the indexing.
4. There is a considerable difference in the orientation measurements between the EBSD and SPED techniques.
5. Ma et al. APL 106 (2015) 101603 shows non-equilibrium oxidation states, Zr¹⁺, Zr²⁺, and Zr³⁺ for suboxide (ZrO_x) rather than just ZrO.
6. Will the initial oxidation product of Zr be amorphous and then form nanocrystal grains?
7. Could the authors calculate the lattice mismatch and interfacial energy for different orientations? In reality, <9% lattice mismatch is needed for obtaining epitaxy. "epitaxial relationship" could be misleading.
8. Some typos: line 210: Fig 2C, line 349: 10 mm etc.

We would like to thank the reviewers for their useful feedback, which have been carefully considered and addressed in the revised manuscript. Please see below responses to individual comments, where the relevant part in the manuscript has been given via the line numbers in the revised document. The changes in the manuscript have been highlighted.

Reviewer #1 remarks and authors' responses

“Detailed TEM examination was carried out on two TEM specimens lifted out of two oxidized Zr grains. It is a long way to expect that these provide representatives for electrochemical corrosion of all alloys in all harsh environments. There are some good insights however presentation could be much better. There appears to be confusion between preferential growth of some oxide grow directions and transformations between oxide types and Zr metal. A major effort to explain the research would be useful. The present version is extremely difficult to understand, and maybe even incomprehensible to a highly trained researcher in the field. The abstract should present main findings and put them into context. There are no conclusions.

Incomprehensible, needs to be translated into language that is understandable to the trained practitioner in oxidation studies.”

We have provided a clearer explanation of the research and made it more accessible by rewriting the abstract and the introductory section of the main text as well as making additions to the results, discussion, and methods sections. In particular, the importance of and the difficulties in studying complex oxide microstructures to understand corrosion properties have been pointed out in the first paragraph in lines 38-52. Furthermore, extensive details of the oxidation of Zr alloys are given in lines 102-120, which explain the mechanism of formation of a protective oxide film. Some of the initial small oxide grains with a particular crystallographic orientation grow preferentially so that the growth stresses in the oxide film are minimised. During this process, the local conditions that stabilised the metastable phases change, e.g. there is a decrease in the stress and the grain size increases, and so a phase transformation to the stable oxide phase is observed.

Edits to the abstract highlight the main aims and findings of the work, namely the ability to study a complex metal-oxide microstructure, such as that of the oxide layer that forms on Zr alloys during waterside corrosion, and to gain a new understanding of the competing mechanisms, the epitaxial strain and the growth stress, that control the formation of a protective oxide film and hence oxidation. These findings could be used to create tailored metal crystallographic textures so that more protective oxides are formed.

While the journal format does not include a “conclusions” section, concluding remarks that summarise the main findings of this manuscript have been extended. We have shown that in alloys which experience inward corrosion and a Pilling-Bedworth ratio of more than one, the epitaxial strain forms a less protective and disordered oxide grain morphology, whereas the growth stress drives a well-ordered protective oxide microstructure, and therefore better corrosion performance. Therefore, it is important to consider local substrate orientations even for materials considered to be highly textured as minor changes in orientation can still lead to significant changes to oxide protectiveness. These findings demonstrate the potential of optimising the processing routes of engineering alloys, based on a crystallographic texture, which is optimum for the corrosion properties. These concluding remarks can be found in lines 578-587.

“a. Please number pages!”

Page numbers have been added to the manuscript.

“1. The problem definition is vague. Harsh environment. Is this molten salt at 1000 C, salt water, or CO₂ + H₂S at 500 C. Please be a little more specific.

2. Is this electrochemical corrosion in an aqueous NaCl solution or a molten salt?”

Details about the corrosion environment used in the present study have been added to the introductory section (lines 126-127) and expanded in the methods section (lines 593-596): “The coupon was subjected to corrosion testing for 46 days at 350°C ± 0.5°C in 316H stainless steel autoclaves in simulated pressurised water reactor (PWR) chemistry at a raised pH level as part of a large-scale testing program⁶⁹, forming an average oxide thickness of ~1.2 µm as estimated from weight gain data”.

“3. It would be useful to have a clear statement of research aims to replace the long discussion at the end of the Main Text.”

The research aims have been made clearer in lines 121-135, to reflect the main aim of demonstrating how our multi-scale analysis can be used to provide a new insight into the oxidation of zirconium alloys in typical pressurised water reactor chemistry by studying the substrate and oxide microstructure and the main driving mechanisms, the epitaxial strain and the growth stress.

“4. Full chemical composition should be given.”

The full composition has been added in the methods section and materials selection subsection, in particular 1.5 wt% Sn, 0.14 wt% Fe, 0.1 wt% Cr and 0.06 wt% Ni, where the balance is Zr (please see lines 591-593).

“5. Justification is needed that weight gain is a useful method to determine passive film thickness, and that there is no oxide loss during corrosion.”

The explanation for using the weight gain as a valid method for determining the oxide film thickness has been added in the methods section and materials selection subsection, including a reference: “Zirconium forms a protective adherent oxide film, where all the oxygen in the chemical reaction produces zirconium oxide, therefore, the weight gain of the specimens is widely used as a direct measure of the oxide film thickness^{68, 71}. “(please see lines 596-599).

“6. Please explain all acronyms. What is SPD?”

The ‘SPED’ acronym (‘SPD’ acronym is not present in the manuscript) and all other acronyms have been defined at the first instance of their use, e.g. line 123. Furthermore, the acronyms have been reexplained in new sections to help with understanding and remove any possible confusion, for instance lines 47, 122, 141, 270, 313, 326, 379, 639.

“7. This work is based on a 2 mm x 2 mm specimen that was oxidised, had most of the oxide removed by grinding, from which two TEM specimens were produced and examined. How representative are these? Justification is needed.”

The electron backscatter diffraction (EBSD) analysis of the metal confirms that the metal grains 1 and 1’ have crystallographic orientations within the expected crystallographic texture measured in single-phase Zr alloys such as the alloy used in this study, Zircaloy-2. Metal grains 2 and 2’ are representative of a smaller but significant (about 20%) fraction of grains that these alloys

contain. (Please see added discussion in lines 174-182). The EBSD analysis of the oxide is based on a very large number of grains (approximately 560,000) and confirms numerous studies^{54, 64}, which show monoclinic oxide texture $\{10\bar{l}\}$ (where $l = 2, 3, 4$) and tetragonal oxide texture $\{001\}$ in the metal grain with typical texture (i.e. grains 1 and 1'). So, although TEM specimens are small, the results agree very well with the EBSD analysis from a much larger number of grains, which agree well with previous studies of this alloy, and so we have high confidence in our results. (Please see lines 147-157, 497-508). The importance of understanding the relationship of very localised measurements provided by state-of-the-art techniques to the overall corrosion behaviour is in-fact an important point this work is trying to highlight.

Furthermore, we have considered the texture of an oxide formed from a large number of metal grains in a typical sample with split-basal texture. We have shown that the modelled oxide macrotexture using the derived orientation-relationships agrees well with macrotexture measurements in the literature. Please see Fig. 9 and the discussion in lines 561-577.

“8. These oxides are thick compared with passive films, 1-2 nm in thickness.”

Further explanations have been added regarding the zirconium oxide film thickness and its protectiveness (lines 102-120), which is also summarised here. Zirconium is a very reactive metal which forms a semi-passivating oxide layer and exhibits cyclic corrosion kinetics. The oxide layer is protective up to a thickness of about 2 microns, at which point it becomes unstable and a breakdown in the protectiveness of the oxide is observed. This leads to the rapid growth of a new fresh protective oxide layer and so the process repeats in a cyclic manner. In this study, we are focused on the first 1.2 μm , and so only on the protective pre-transition oxide, which has been clarified in lines 142-144.

“9. Oxide growth direction is undefined.”

The oxide growth direction has now been marked on several figures in order avoid any confusion: Fig. 1c, Fig. 4a, Fig. 5a, Fig. 6a and Fig. 7. The inward oxide growth has been explained and discussed in lines 86-108.

“10. the transformation from Zr to ZrO₂ is presumably by oxidation, If so please state this. What is the evidence for the transformation of tetragonal ZrO₂ to monoclinic ZrO₂, and how does this occur?”

Yes, the transformation of Zr to ZrO₂ is by oxidation, which has been made clearer in lines 102-110. The tetragonal ZrO₂ is meta-stable at typical reactor conditions (please see lines 119-122 and 157-163). Further explanation and extensive literature evidence for the tetragonal ZrO₂ to monoclinic ZrO₂ transformations have been added in lines 110-120. The tetragonal phase is stabilised by a combination of small grain size, compressive stress and oxygen vacancies, all present near the metal-oxide interface. As the oxide layer grows inwards, the older tetragonal grains get further away from the metal-oxide interfaces where the compressive stress decreases, and they grow larger, and so the conditions that stabilised them are no longer present, and they transform to the stable monoclinic phase.

“11. Or are you trying to say that Zr oxidised to both oxide variants independently? Or are you trying to say that one oxide variant grows on top of the other? What is the evidence for any of this?”

There has been extensive evidence that Zr oxidises to a mixture of multiple oxide phases (hexagonal ZrO, tetragonal ZrO₂ and monoclinic ZrO₂), i.e. with different crystal structure, as discussed in lines 102-120. As some of these oxide phases are metastable (stabilised by local factors

near the metal-oxide interface), as the oxidation progresses and the stabilising factors change, they transform to the stable oxide phase (monoclinic). In this study, for the first time we show evidence that during the oxidation of Zr metal grain with an orientation from the typical alloy texture the tetragonal ZrO₂ phase forms first, which then transforms to monoclinic ZrO₂ as the oxide thickens and is found to be beneficial for the protectiveness of the oxide (lines 218-231, 448-464, 497-512). The evidence for this comes from the measured orientation data for the two phases using electron backscatter diffraction in Fig. 3. There has been much debate in the literature as to whether tetragonal oxide is always formed as a precursor to monoclinic, or if both can form independently and this study is the first to show direct evidence of this. As discussed in lines 218-230, our model of possible theoretical orientation relationships shows that the experimentally measured monoclinic texture variants could only be formed if they had formed as tetragonal ZrO₂ first and then transformed to monoclinic based on the identified orientation relationship. Furthermore, an explanation has been added in lines 235-237 to clarify what a crystallographic symmetry variant is: it refers to a crystal with the same crystal structure as another variant, but with different orientation with respect to the parent crystal from which it formed during a phase transformation.

“12. What is precession diffraction? Please explain. Presumably the black areas are where there is lack of data?”

Please see explanation added in lines 655-659, including a reference for further information, and further specifications regarding the scanning precession electron diffraction technique, including orientation and phase reliability index criteria for coloured and black areas in the maps in lines 663 and 664. The reliability index criteria are also given in the relevant figure captions (Fig. 4 and 5).

“13. It would be good to indicate the orientations of the oxide grains and of the Zr grains in Fig. 4(a)”

The orientation of the metal grain has been denoted by superimposing the Zr unit cell in Figs. 4a and 5a. The monoclinic oxide orientations are given in Fig. 4c and d.

“14. The majority of the gains in Fig. 5b are black which presumably means unidentified.”

The procedure for identifying grains as indexed (coloured) or non-indexed ('black') is described in the methods section, the scanning precession electron diffraction subsection and appropriate references for this established technique are given in lines 663 and 664. The reliability index is chosen so that we have a very high confidence in the grains that are indexed even if there is a high fraction of non-indexed grains, since we are using the electron backscatter diffraction results from a much larger number of oxide grains to compare and verify with these from the scanning precession electron diffraction.

“15. The model in Fig. 7 is neat. However, please indicate the confidence level, as most of grains were black in Fig. 5.”

There is a high level of confidence in the model presented in Fig. 7, because although as you point out there is a large number of nonindexed grains using the SPED technique, we have achieved a very high indexing confidence in the EBSD results, which allowed us to sample more than 5 million oxide grains.

“16. The presented discussion about water splitting can only be relevant until the first oxide monolayer forms.”

The difference in the energetics of the interactions of oxygen and hydrogen with different zirconium metal orientations is relevant for the oxidation process beyond the first oxide monolayer, because the oxide layer grows inwards by diffusion of oxygen into the metal after it diffuses through the existing oxide. Zr has a high O solubility and so there is a layer of oxygen saturated metal followed by layer of a combination of suboxide and metastable oxide phase, which are more likely to form in the presence of higher fraction of oxygen vacancies. And there would be a higher fraction of oxygen vacancies for metal orientations with faster and deeper oxygen penetration, as already described in lines 438-447.

“17. What is the orientation of a Zr grain with a split basal orientation? There appears to be a mixup of macroscopic and gain focused conceptions.”

That is a Zr grain with the basal pole $\langle 0002 \rangle$ of the hexagonal close packed crystal inclined at an angle with respect to the normal direction, i.e. the direction normal to the metal-oxide interface, within the range for a ‘split-basal’ texture in single-phase Zr alloys. The “‘split-basal’ orientation” was used as a short way to describe these grains, but this has now been removed and explained better. Please see lines 429-431 and 449-450. As described in lines 408-412-442, “in a typical metal grain in split-basal textured single-phase Zr alloys, the c-axis of the hcp crystal is positioned at 20° to 40° away from the outer surface normal, and so pyramidal planes with Miller indices $\{h0il\}$ are close to parallel to the outer surface (Fig. 7a)”

“18. There is also confusing terminology about oxide transformation, when the model indicates growth of grains of particular orientations on other grains.”

There is “growth of grains of particular orientation on other grains”, which is the effect of an epitaxial strain, or in other words, lattice matching between two crystal structures. We show evidence for the epitaxial strain as a mechanism in both the metal-oxide transformation and the transformation between oxide phases (lines 218-230, 246-257, 363-376). New explanations have been added in the introductory part as mentioned previously to make clearer the subject of the different types of transformations and why they occur (lines 102-120).

“19. The atomic relations in Fig. 8 are instructive. However, the split-basal orientation needs to be included.”

As mentioned in the response to comment 17, the ‘split-basal’ orientation refers to the orientation of the Zr substrate grain with the basal pole at an angle with respect to the cladding tube surface. On the other hand, Fig. 8 shows the atomic interfaces formed between the different crystal phases and so it demonstrates the relative atom positions. Fig. 8 has been extended to include schematics of the oxide microstructure and the normal direction in order to make clearer the relative orientation between the atomic interfaces and the global orientations.

“20. Are the strains in Table 1 relevant. If so please provide TEM plane spacings of these oxides in support”

The quantities shown in Table 1 have been renamed from ‘strains’ to ‘mismatch’ to reflect the fact that these represent theoretical differences in the Zr-Zr distances in the corresponding crystal structure. These are used to as an approximate measure of the epitaxial strains in order to compare qualitatively the lattice restrictions during the phase transformations in both regions, as explained in lines 465-471.

Reviewer #2 remarks and authors' responses

“The study investigates the mechanisms contributing to zirconium oxide phase formation during corrosion of zirconium. The research method utilized orientation mapping techniques in scanning electron microscope and transmission electron microscope to determine the texture in parent and oxide phases. The experimental data and the theoretical calculations orientation relationship in the phases reveal the energetic favor of the tetragonal oxide phase nucleation on zirconium substrate during corrosion. This is the first time that a study demonstrates the possibility of tetragonal oxide nucleation in zirconium though such theories have been discussed. The manuscript argues that the orientation of the zirconium substrate would affect the oxide nucleation and growth process. Thus, the research work argues the influence of epitaxial strain and growth stress in the formation of oxide phases and their texture in zirconium.

The following are the areas where more clarity and information are needed.

1. The authors' conclusion on the nucleation of the tetragonal oxide phases is based on the post-corrosion analysis of the microstructure and assumed orientation relationship calculations between Zr, tetragonal oxide, and monoclinic oxide. The analyzed sample has more than 95% transformed monoclinic oxide phases. Indeed, the results are high-quality. However, the authors may consider using in situ TEM observation of such tetragonal oxide nucleation can provide direct and more convincing evidence. Another option is to have an oxidation condition under which the tetragonal phase is a major phase. There are several papers regarding using precession electron diffraction to study the oxidation of Zr or zircaloy and the tetragonal phase is popular (J. Nucl. Mater. 556 (2021) 153196; Scr Mater. 145 (2018) 95).”

We thank the reviewer for this comment on the quality of the results. We would like to emphasise the importance of the compressive stresses in the oxide layer in stabilising the tetragonal phase, as discussed in lines 102-120. Therefore, as described in the manuscript (lines 47-52), TEM-based techniques are performed on electron transparent samples, which alter the microstructure by the partial release of these stresses and the transformation of the stress-stabilised tetragonal grains. For that reason, we do not consider in-situ TEM as a viable option to give insights into the effect of the compressive stresses on the tetragonal oxide nucleation. As Harlow et al. points out (Scr Mater. 145 (2018) 95) significant care must be taken when interpreting TEM results for the tetragonal phase, as some stress release is always present. We also note that we used scanning precession diffraction (SPED) in the TEM to study the oxide thickness, grain morphology and to verify the monoclinic oxide texture that was measured using EBSD, where the stress state is maintained. There are useful applications of using SPED for the study of the tetragonal phase, as the reviewer has pointed out, such as when it is chemically stabilised in the case of J. Nucl. Mater. 556 (2021) 153196, or when it is stabilised by the small grain size. However, there are numerous results from non-destructive techniques such as XRD⁴⁸⁻⁵¹ showing that the monoclinic phase is the major phase in standard autoclave-formed oxides, although some transformation of the minor tetragonal phase is expected during TEM preparation.

“2. In region 2, the number of tetragonal oxide grains analyzed is ~530, which is 3-4 times lesser than that in region 1. The pole figures in region 2 show close to random texture. How representative is the pole figure calculated for the phase in region 2 from a smaller number of grains?”

Although there are a smaller number of tetragonal grains in region 2, we consider the result of a random texture reliable. We have confirmed the results for both regions 1 and 2, via comparison with regions 1' and 2', which contain 258 and 285 tetragonal grains respectively, as

shown in the Supplementary material. As the effect of the epitaxial strain on the tetragonal texture has been captured with only 258 grains in region 1', we consider the 531 tetragonal grains in region 2 to be reliable for the determination of a random texture. The manuscript has been edited to include this additional argument in lines 199-202.

"3. Since each oxide grain has one orientation data point in the SEM-EBSD study, it is critical to comment on the correctness of the indexing."

Due to the fine grain size of the oxide, EBSD must be performed using a high-resolution SEM instrument. This retains sufficient brightness in a small beam footprint, which reduces overlapping of diffraction patterns from neighbouring grains and so improves indexing. It is thought that the relatively low indexing rate in the oxide here (~60 %) results from overlapping patterns. It is therefore likely that when a pattern is indexed, it is from a position where the beam coincides towards the centre of an oxide grain, reducing the effect of overlapping patterns from neighbouring grains. We therefore have high confidence in the indexing in these regions, both in terms of orientation accuracy (Average mean angular deviation (MAD) of 1.07 and 1.45 for the monoclinic and tetragonal phases respectively), and in the phase accuracy, as the monoclinic and tetragonal phases have significantly different crystal structures, and so misindexing is unlikely. The accuracy of the indexing using bulk EBSD is reflected by the similarities in observed textures to those observed by non-destructive XRD measurements in the literature⁴⁸⁻⁵¹. The discussion has been added to the manuscript in lines 147-157.

"4. There is a considerable difference in the orientation measurements between the EBSD and SPED techniques."

The difference in the EBSD and SPED monoclinic oxide texture measurements have been carefully considered and the discussion has been expanded in lines 287-298 and 353-358. Although there is a difference between the EBSD and SPED monoclinic oxide texture measurements (15° and 5° difference in the peak misorientation angle away from the {10-6} monoclinic pole and the {11-2} monoclinic pole in region 1 and region 2, respectively), that difference does not change the proposed mechanisms for monoclinic oxide texture formation. It can be clearly seen in Figs. 4d and 5d that the SPED measured {10-6} and the {11-2} pole figures in regions 1 and 2 confirm the main patterns shown in the EBSD measured pole figures in Fig. 3. Moreover, the difference between the maximum in the misorientation angle distributions and the theoretical value is about 13° for both the SPED and the EBSD in region 1, and 4° and 7° for, respectively, SPED and EBSD in region 2. These values are within the expected uncertainties, and further confirm a very good agreement between the two techniques and between the experimental measurements and the theoretical model. The main reasons for the differences between the two techniques include the large difference between the sampling statistics in the two techniques, misalignment between the samples used in each technique, internal misorientations within the metal grain, misindexing of some SPED data and relative surface orientation before/after oxide removal. Additionally, there are differences in the measured grain population between the two techniques with EBSD biased towards larger grains. For these reasons, it is important to combine different techniques when studying such complex oxide systems.

"5. Ma et al. APL 106 (2015) 101603 shows non-equilibrium oxidation states, Zr1+, Zr2+, and Zr3+ for suboxide (ZrOx) rather than just ZrO."

We thank the reviewer for the insightful reference, which finds that a gradual change in the stoichiometry through the formation of suboxides is energetically more favourable compared to an

abrupt change from Zr to ZrO₂ in the initial stages of the oxidation. This supports the findings in the current manuscript, that the metastable phases provide a more gradual route for the formation of monoclinic ZrO₂ and therefore a more protective oxide layer. Discussion and reference have been added in lines 520-528, 532-533.

“6. Will the initial oxidation product of Zr be amorphous and then form nanocrystal grains?”

There have been some limited suggestions in the literature for the presence of amorphous grains in the initial oxidation stages of Zr (e.g. Ploc and Zhou et al.), however, the general opinion is that if they form, they only form in the very initial stage of the oxidation and certainly would be of limited number. Furthermore, the effects of the epitaxial strains that we found are strong enough to support the idea that the amorphous phase might only form in very limited oxide grains and on very short time and length scales in single-phase Zr alloys.

Ploc R. A. Transmission electron microscopy of thin (<2000 Å) thermally formed ZrO₂ films. J Nucl Mater 1968; 28:48–60.

Zhou BX, Li Q, Yao MY, Liu WQ, Chu YL. Effect of water chemistry and composition on microstructural evolution of oxide on Zr-alloys. J ASTM Int 2009:360.

“7. Could the authors calculate the lattice mismatch and interfacial energy for different orientations? In reality, <9% lattice mismatch is needed for obtaining epitaxy. “epitaxial relationship” could be misleading.”

The theoretical lattice mismatch has been calculated in Table 1, which are based on the Zr-Zr distances in the corresponding lattices. To improve clarity, Table 1 and the relevant discussion in lines 466-471 have been updated. These measures are only used to qualitatively compare the mismatch in the transformations in the two regions, as in reality there will be atom relaxations, specifically in the normal directions and so the mismatch would be lower, so it would fall within the quoted number of 9%. The calculation of the interfacial energies is beyond the aims of the current manuscript, although it would be an interesting study. Our aim was to show that our crystallographic texture measurements clearly showed the presence of orientation relationships between the metal and oxide phases, and we have defined these by modelling the theoretical orientation relationships.

“8. Some typos: line 210: Fig 2C, line 349: 10 mm etc.”

The typos have been corrected.

REVIEWERS' COMMENTS

Reviewer #1 (Remarks to the Author):

nice work, publish, reviewers' points dealt with

Reviewer #3 (Remarks to the Author):

I have read the revised manuscript and I am satisfied with the changes and how the authors addressed the points raised.